# Reversal of ATP synthase is a key attribute accompanying cellular differentiation of *Trypanosoma brucei* insect forms

Michaela Kunzová [1,2], Eva Doleželová[1], Martin Moos [3], Brian Panicucci[1] & Alena Zíková [1,2] ✉

The mitochondrial $F_oF_1$-ATP synthase is a reversible nanomachine that normally produces ATP via oxidative phosphorylation but under stress conditions it can reverse to maintain the mitochondrial membrane potential at the expense of ATP, a process regulated by the conserved inhibitory factor 1 (IF1). We show that ATP synthase reversal also occurs during in vitro-induced differentiation of the unicellular parasite *Trypanosoma brucei*, partially mirroring events in the tsetse fly. Differentiation of insect forms is marked by increased expression of alternative oxidase and reduced levels of trypanosomal IF1 (TbIF1), changes that may promote ATP synthase reversal. Parasites lacking TbIF1 efficiently progressed to the mammalian-infective form, coinciding with increased ATP synthase reversal, a higher ADP/ATP ratio, elevated phosphorylation of AMP-activated protein kinase (AMPK), enhanced proline-supported respiration, and increased mitochondrial and cellular reactive oxygen species (ROS). In contrast, inducible TbIF1 overexpression diminished these hallmarks and locked parasites in the initial insect stage. Our findings reveal that TbIF1 downregulation enables life cycle progression and underscore a regulatory role for the ATP synthase–IF1 axis.

Mitochondria are multifaceted organelles that play a crucial role in cellular bioenergetics, metabolism, and intracellular signaling[1,2]. Central to these processes is the $F_oF_1$ ATP synthase (ATP synthase), a molecular nanomachine that synthesizes the majority of cellular ATP via oxidative phosphorylation (OXPHOS) under aerobic conditions[3,4]. The synthesizing, or forward, mode of this complex is driven by the high proton motive force (pmf) generated by the electron transport chain (ETC) complexes. As with most enzymes, the complex is reversible, and thus can assume a hydrolytic function when the pmf is compromised (e.g., during hypoxia/anoxia conditions, in cells with dysfunctional ETC, or with low electron input)[5,6]. This hydrolytic activity consumes large amounts of ATP, making it potentially harmful to the cell if not properly regulated. In mitochondria, a unique mechanism involving a short polypeptide named Inhibitory Factor 1 (IF1) has evolved to control the hydrolytic mode of this rotatory machine[7]. The presence of IF1 in almost all aerobic eukaryotes with typical mitochondria suggests that the mechanism regulating ATP synthase activity is conserved among various groups of organisms[8].

The unique characteristic of IF1 is its ability to selectively inhibit the hydrolytic activity of ATP synthase, acting as a unidirectional inhibitor[9]. This inhibition is influenced by environmental conditions such as a pH below 7 and high concentrations of cations (e.g., $Ca^{2+}$); both of these

conditions activate IF1[10,11]. The process of IF1 activation involves the exposure of its intrinsically disordered N-terminus, leading to the formation of a short α-helix which then protrudes into the $F_1$ moiety of ATP synthase. It forms contacts with the α-helical coiled-coil region of the $F_1$-ATPase subunit γ, thereby halting its clockwise rotation (as viewed perpendicular to the mitochondrial membrane plane, with the mitochondrial matrix on top)[12]. When the pmf is restored, mitochondrial membrane potential (ΔΨm) drives the clockwise rotation of the subunit γ, which forcefully ejects IF1 from the $F_1$ domain[13]. Notably, it has been proposed that IF1 also suppresses the ATP synthetic activity of the complex in certain human cells[14,15], although this view has been recently challenged by cell-based and in vitro assays and hence remains uncertain[9,16,17].

There are many biological roles for IF1 that have been so far described. For example, IF1 plays a key role in inhibiting intermediate assemblies of $F_1$ particles during complex assembly[18,19], or in preserving the cellular ATP pool during ischemic conditions in cardiac tissues[20]. Another significant function of IF1 is its association with the type I dimer of the mammalian ATP synthase complex[21], which can self-assemble into longitudinal rows decorating the rims of lamellar cristae[22,23]. By binding to ATP synthase and stabilizing the oligomers in these long rows, IF1 levels influence the mitochondrial cristae structure[24–27]. Recently, the pro-oncogenic potential of IF1

[1]Institute of Parasitology, Biology Centre, Czech Academy of Sciences, Ceske Budejovice, Czech Republic. [2]Faculty of Science, University of South Bohemia, Ceske Budejovice, Czech Republic. [3]Institute of Entomology, Biology Centre, Czech Academy of Sciences, Ceske Budejovice, Czech Republic. ✉e-mail: azikova@paru.cas.cz

has garnered considerable attention, as IF1 expression is elevated in highly proliferative cancer cells and its presence promotes proliferation and resistance to hypoxic conditions and cell death[28]. Other potential mechanisms of IF1 action are relevant to mitophagy, neurodegenerative diseases and aging[16]. Last, but not least, IF1 might be necessary not only under pathophysiological conditions, but also under normal physiological conditions. Accumulating evidence supports the possibility that both ATP synthase activities co-exist within a single coupled intact mitochondrion[29–32]. This underscores the importance of the highly conserved IF1, whose activity might be needed to block ATP hydrolysis under OXPHOS conditions.

In this study, we investigated the role of IF1 during the extensive metabolic reprogramming that underlies the in vitro-induced cellular differentiation of the insect forms of the unicellular parasite *Trypanosoma brucei*[33]. *T. brucei* belongs to the group of African trypanosomes, which impose significant medical and economic burdens as infectious agents of humans and domestic animals, causing Human and Animal African Trypanosomiasis, respectively[34,35]. *T. brucei* is a digenetic parasite that alternates between a mammalian host and an insect vector, the tsetse fly. The complex life cycle of these parasites involves a programmed developmental progression through various life cycle stages, each characterized by distinct expression profiles reflecting the parasite´s specific environments[36,37].

In the bloodstream of the mammalian host, the bloodstream form of the parasite is covered by a dense variant surface glycoprotein coat[38], and its metabolism is heavily dependent on glucose oxidation via glycolysis[39]. The mitochondrion of this parasite is reduced in size and activity, yet it retains essential functions, for example, by maintaining the glycolytic redox balance through the dihydroxyacetone/glycerol-3-phosphate shuttle coupled to the reduction of oxygen via an alternative oxidase (AOX)[40]. In addition, the canonical cytochrome-mediated ETC is absent, and the $\Delta\Psi$m necessary for protein import and ion homeostasis is maintained by ATP synthase operating in reverse[41,42]. The ATP demands are met by either the supply from the cytosol or the ATP can also be generated in the mitochondrial matrix by substrate phosphorylation[43,44]. The expression of *T. brucei* IF1 (TbIF1) is fully restricted in the bloodstream forms, and experimentally induced TbIF1 expression is lethal to the parasite[45].

Upon ingestion by the tsetse fly, the bloodstream form parasite rapidly differentiates into the procyclic form, which populates the fly's midgut. This form is characterized initially by a glycoprotein coat composed of GPEET and EP procyclins, which are subsequently replaced only by EP procyclin[46]. The procyclic form fully utilizes the abundant amino acids, such as proline and threonine, oxidizing them through mitochondrial metabolism via a partial tricarboxylic acid (TCA) cycle[47]. The AOX expression is downregulated, and the majority of electrons released from amino acid oxidation are transferred to oxygen via ETC complexes III and IV, generating the pmf. In this life cycle stage, ATP synthase operates in its canonical mode, synthesizing ATP[48,49]. In addition to OXPHOS, cellular ATP is also generated by powerful mitochondrial substrate phosphorylation involving the ATP-generating succinyl-CoA synthetase, which provides the parasite with flexibility in terms of energy metabolism[33,50]. Procyclic form and bloodstream form parasites are the only two forms commonly grown as proliferating cultures in the laboratory.

Once the infection is fully established in the midgut, the parasite migrates to the salivary glands via proventriculus, transitioning into the epimastigote form, which is characterized by a coat composed of different GPI-anchored proteins called brucei-alanine-rich-proteins (BARPs)[51]. The epimastigote form inhabits the salivary glands and later transforms into the quiescent, cell-cycle arrested metacyclic parasites, which are infectious to the mammalian host. The metacyclic form is pre-adapted for the survival in the mammalian host by adjusted gene expression profile[52,53]. While the metabolism of the latter two forms remains to be fully elucidated, partial understanding can be derived from studies utilizing in vitro differentiation that mimics, to some extent, the developmental processes occurring in the tsetse fly[54,55]. This differentiation depends on the over-expression of RNA-binding protein 6 (RBP6), whose expression is highly upregulated in salivary gland parasites[56] and, in vitro, it triggers the transformation of procyclic

parasites into epimastigotes and metacyclics in the process called metacyclogenesis[57,58]. However, the in vitro-generated epimastigotes likely correspond only to attached (late) epimastigotes and thus this system does not recapitulate the full progression from midgut to proventriculus and salivary glands.

Omics and cell-based analyses have established that during this differentiation, there is a significant remodeling of the ETC, with AOX expression being strongly upregulated while ETC complexes III and IV are downregulated. Moreover, TbIF1 expression is developmentally downregulated, suggesting a need for ATP synthase reversal during parasite differentiation[55,59].

Gain- and loss-of-function experiments revealed that TbIF1 depletion promotes in vitro differentiation from the procyclic to the metacyclic form, while its inducible overexpression blocks the early steps of this process. We propose that the absence of TbIF1 allows the reversal of the partial ATP synthase pool promoted by the increased expression of AOX. This is accompanied by changes in cellular respiration, reactive oxygen species (ROS) levels, ADP/ATP ratio, and activation of AMP-activated protein kinase (AMPK), an enzyme that acts as an energy sensor in cells. TbIF1 programmed down-regulation is a key adaptive mechanism that enables the ATP synthase reversal and, as a consequence, TbIF1 modulates *Trypanosoma* energy metabolism during cellular differentiation of the insect forms.

## Results
### Levels of TbIF1 regulates the RBP6 induced progression through *T. brucei* development

Given that the IF1 is a master regulator of mitochondrial physiology and that levels of TbIF1 expression are downregulated during the development in the tsetse fly[37] as well as following the RBP6 induction (Fig. 1A, left panel)[55], we were interested in its role in the parasite differentiation. To this end, we generated procyclic cell lines in which TbIF1 was either lacking or inducible expressed at levels above its endogenous expression (Fig. 1A). To generate a cell line lacking TbIF1, we proceeded to remove both TbIF1 alleles by homologous recombination. This was followed by the verification of the successful replacement of both alleles with cassettes containing T7 RNAP and TetR by PCR (Supplementary Information Fig. 1). Subsequently, a cassette containing the RBP6 gene under the control of T7 RNAP and TetR was introduced to allow for the tetracycline inducible expression of RBP6 (RBP6$^{OE}$_TbIF1dKO). To generate a cell line with high expression of TbIF1 during RBP6-induced differentiation, a cassette ensuring tetracycline inducible overexpression of TbIF1 was transfected into the RBP6$^{OE}$ cell line (RBP6$^{OE}$_TbIF1$^{OE}$). The RBP6-driven differentiation of the generated cell lines was induced by tetracycline. The induced RBP6 and TbIF1 expression was verified by Western blot. As anticipated, TbIF1 expression was undetectable in RBP6$^{OE}$_TbIF1dKO, while it remained consistently elevated in RBP6$^{OE}$_TbIF1$^{OE}$ due to the constant presence of tetracycline in the medium (Fig. 1A, middle and right panels).

A typical manifestation of RBP6-induced differentiation is a slowdown in cell growth, as the parasite differentiates from dividing procyclic form and epimastigotes to the metacyclic form that is arrested in the G1/G0 cell cycle phase[58]. The RBP6$^{OE}$_TbIF1dKO line exhibited a more pronounced growth defect compared to the RBP6$^{OE}$ cell line, whereas the RBP6$^{OE}$_TbIF1$^{OE}$ cell line showed the least severe growth defect (Fig. 1B). To gain insight into the dynamics of the differentiation process, we determined the individual life cycle forms (procyclic, epimastigote, and metacyclic). We took into considerations stage-specific morphology cues, such as shape and cell size as well as the relative position of the kinetoplast to the nucleus. Procyclic cells have their mitochondrial DNA (kDNA) located in the posterior part of the cell. During the maturation of epimastigotes, their kDNA migrates, and it is typically located near the nucleus at the anterior part of the cell. In metacyclic trypomastigotes, which are smaller than both procyclics and epimastigotes, the kinetoplast is located at the more rounded posterior tip (Supplementary Information Fig. 2). The RBP6$^{OE}$ showed a typical differentiation profile[55] with almost 80% of epimastigotes on day 2 and with close to 40% of metacyclic parasites at day 6. In accordance with the growth

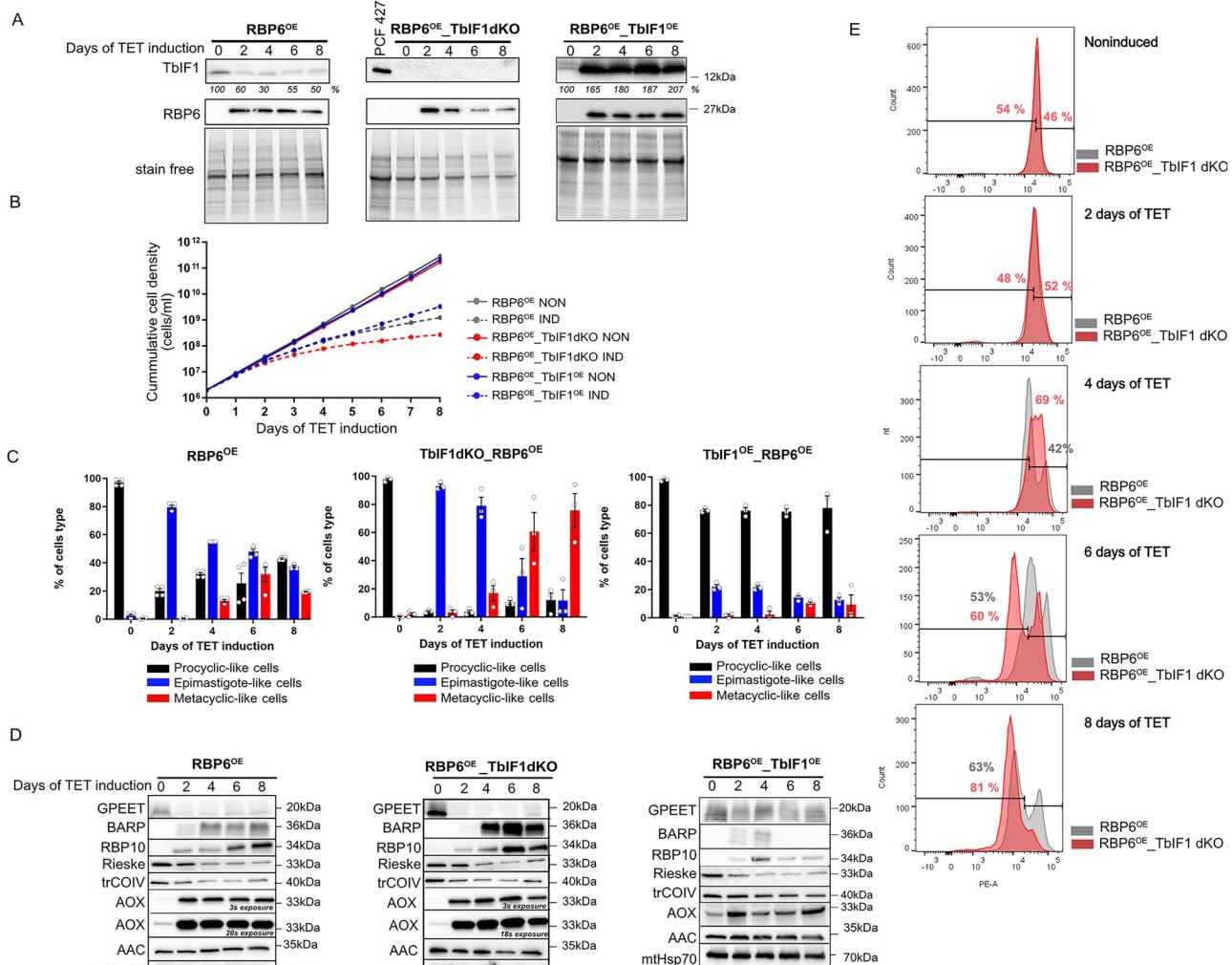

**Fig. 1 | Lack of TbIF1 enhances differentiation of procyclic *T. brucei* cells.**
**A** Western blot analysis of whole cell lysates from the parental RBP6$^{OE}$,
RBP6$^{OE}$_TbIF1dKO and RBP6$^{OE}$_TbIF1$^{OE}$ cells harvested at timepoints 0, 2, 4, 6, and
8 after the tetracycline (TET)-induced RBP6 overexpression. The procyclic strain
PCF 427 we used as a control for IF1 detection. Stain free gels served as a loading
control for equal loading of $1 \times 10^7$ cells/well. The numbers beneath the blots
represent the abundance of immunodetected protein expressed as a percentage of
the noninduced sample after normalizing to the signal intensity of the loading

control. **B** Growth curves of RBP6$^{OE}$, RBP6$^{OE}$_TbIF1dKO, and RBP6$^{OE}$_TbIF1$^{OE}$
measured for 8 days. **C** Cell type scoring analysis for the presence procyclic-like of
epimastigote-like, and metacyclic-like cells upon induction of RBP6 overexpression
in all three cell lines. **D** Western blot analysis of whole cell lysates from RBP6$^{OE}$,
RBP6$^{OE}$_TbIF1dKO, and RBP6$^{OE}$_TbIF1$^{OE}$ cells undergoing differentiation using a
panel of various antibodies. **E** Flow cytometry overlay histograms for TMRE-stained
cell population of RBP6$^{OE}$ and RBP6$^{OE}$_TbIF1dKO noninduced and induced for 2, 4,
6, and 8 days.

curves, RBP6$^{OE}$_TbIF1dKO demonstrated enhanced differentiation effi-
ciency, with the detection of 60–80% metacyclic trypomastigotes at days 6
and 8, which was significantly higher than what observed in RBP6$^{OE}$. In
contrast, RBP6$^{OE}$_TbIF1$^{OE}$ exhibited a significant impairment of differ-
entiation to epimastigotes, with less than 20% of the culture reaching this
stage at day 2 post induction (Fig. 1C).

RBP6-induced differentiation is accompanied by a change in the
expression profile, with some marker proteins being readily visualised
using available antibodies. In RBP6$^{OE}$, the expression of the surface gly-
coprotein GPEET is downregulated and later replaced by BARP, which
signals the presence of mature epimastigotes. Additionally, the expression
of RNA binding protein 10 (RBP10), a protein whose expression corre-
lates with that of the bloodstream-form like gene profile[60], is increased
during differentiation. As a consequence of the remodeling of the cano-
nical ETC, the subunits of complexes III and IV, Rieske and trCOIV,
respectively, are downregulated, while the subunit of the AOX is strongly
upregulated shortly after the induction of RBP6 overexpression (Fig. 1D,
left panel). In the case of RBP6$^{OE}$_TbIF1dKO, a more pronounced increase
in BARP and RBP10 signals was observed, which is consistent with the

growth curves indicating a higher proportion of mature epimastigotes and
metacyclics in the culture (Fig. 1D, middle panel). In contrast, for
RBP6$^{OE}$_TbIF1$^{OE}$, there was no decrease in GPEET, and the signal for
BARP was only barely discernible on days 2 and 4 following the induction
of RBP6$^{OE}$. The remaining markers exhibited a comparable trend to that
observed in RBP6$^{OE}$, albeit to a lesser extent. This suggests that RBP6-
driven programmed differentiation was triggered on the molecular level in
the RBP6$^{OE}$_TbIF1$^{OE}$ cells. However, the transition into the next life cycle
stage was prohibited by the TbIF1 overexpression as the procyclic cells
appear to be the dominant cell population in the culture (Fig. 1D,
right panel).

Furthermore, the enhanced differentiation of RBP6$^{OE}$_TbIF1dKO is
also corroborated by the histograms of TMRE-stained cells, which are
employed for the estimation of $\Delta\Psi$m. Figure 1E illustrates the typical pattern
of RBP6$^{OE}$ cells (represented by grey peaks) when a distinct cell population
exhibits increased TMRE fluorescence intensity on days 4 and 6. Inversely,
another distinct cell population exhibits a decrease in $\Delta\Psi$m on day 8
(63 ± 7%). In RBP6$^{OE}$_TbIF1dKO, this pattern is even more pronounced on
day 4, 69 ± 6% of cells exhibit increased TMRE fluorescence, while on day 8,

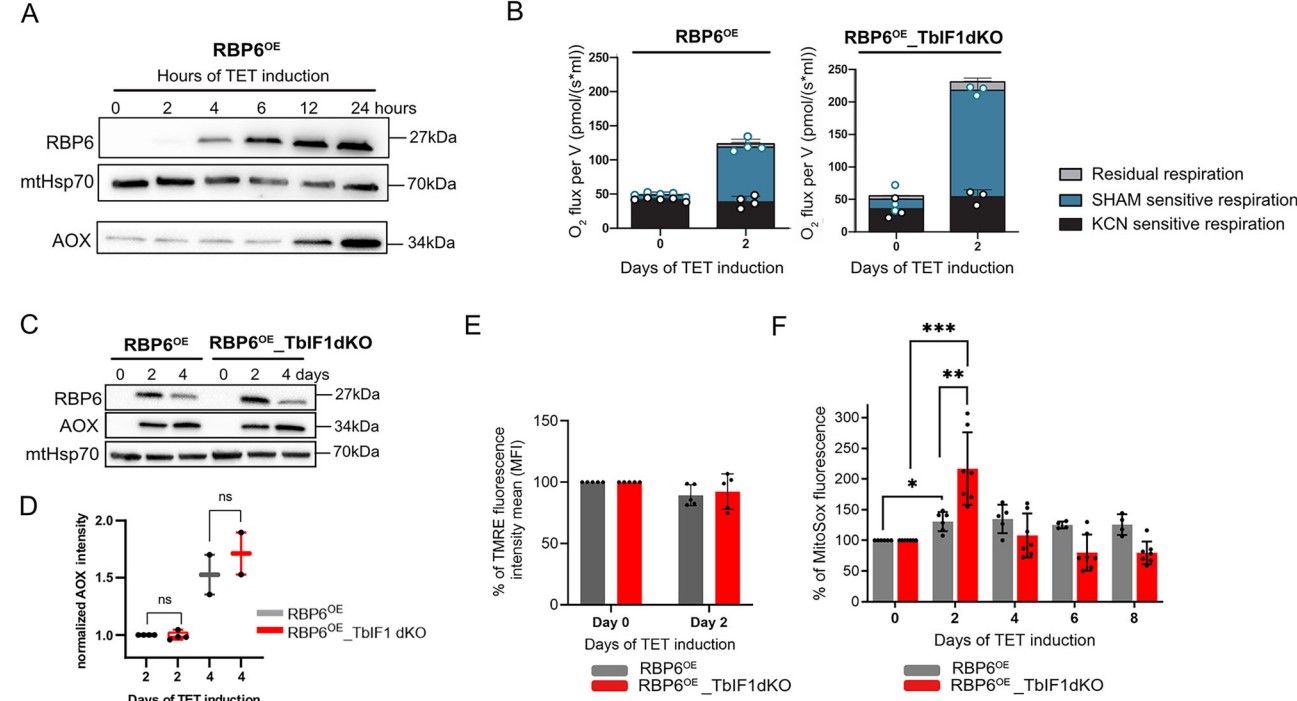

**Fig. 2 | Absence of TbIF1 during *T. brucei* procyclic differentiation enhances oxygen consumption and mROS generation. A** Western blot analysis of whole cell lysates from RBP6^OE cells induced for 0, 2, 4, 6, 12, and 24 h. The immunoblot probed with anti-mitochondrial Hsp70 antibody served as loading control. **B** Oxygen consumption rates expressed as picomoles of oxygen consumed by $2 \times 10^7$ cells per second per milliliter in the presence of 5 mM proline in intact RBP6^OE and RBP6^OE_TbIF1dKO cells induced for 0 and 2 days. The ratio between complex IV- and AOX-mediated respiration was determined using KCN, a potent inhibitor of complex IV, and SHAM, a potent inhibitor of AOX. (means ± s.d., *n* = 3) **C** Western blot analysis of whole-cell lysates from RBP6^OE and RBP6^OE_TbIF1dKO induced for 0, 2 and 4 days using antibodies against RBP6 and AOX. The immunoblot probed with anti-mitochondrial Hsp70 antibody served as loading control. **D** Densitometric quantification of AOX levels in RBP6^OE and RBP6^OE_TbIF1dKO cells induced for 2 and 4 days. The AOX band intensities were normalized to HSP70 and expressed relative to RBP6^OE day 2 levels. The values were analyzed statistically using GraphPad Prism 10.5.0 software (means ± SD, *n* = 2–4, Student's unpaired t-test). **E** Flow cytometry analysis of TMRE-stained RBP6^OE and RBP6^OE_TbIF1dKO cells induced for 2 days. (means ± s.d., *n* = 5) **F** Flow cytometry analysis of MitoSox treated cells to detect mROS levels. Individual values shown as dots. (means ± s.d., *n* = 4-7, * *P* < 0.05, *** *P* < 0.001).

81 ± 8% of cells are present in the population with lower ΔΨm. When these results are combined with cell type scoring (Fig. 1C), it indicates that BARP-expressing epimastigotes exhibit a higher ΔΨm, whereas metacyclics arrested in the cell cycle exhibit a lower ΔΨm.

These findings indicate that changes in TbIF1 levels are closely associated with the progression of RBP6-induced differentiation. The absence of TbIF1 correlates with a higher proportion of metacyclic cells (i.e. enhanced metacyclogenesis), whereas its sustained expression significantly impairs the parasite's capacity to differentiate into epimastigotes, let alone infectious metacyclic form. Given that the primary function of IF1 is to inhibit the reversal of ATP synthase, this activity appears to be an essential attribute during parasite differentiation.

## TbIF1 modulates changes in mitochondrial functions during parasite differentiation

To gain insight into the mechanism by which the absence of TbIF1 allows procyclic cells to progress more efficiently through differentiation and to ascertain the role of ATP synthase in this process, we examined typical mitochondrial functions (i.e., complexes III/IV- and AOX-mediated respiration, mitochondrial reactive oxygen species (mROS) levels, and ΔΨm) in the RBP6^OE_TbIF1dKO cell line in comparison with RBP6^OE. One of the earliest hallmarks of RBP6-induced differentiation is the increased expression of AOX, which is observed as early as 12 h after the tetracycline induction (Fig. 2A). Generally, AOX competes with complex III for electrons from ubiquinol, subsequently transferring them directly to oxygen; although, it does not contribute to the pmf. The elevated levels of AOX were accompanied by increase in proline-based respiration in live intact cells on

day 2 in both cell lines, suggesting that this amino acid is being consumed and oxidized at a higher rate, leading to an increase in the electron flow to the ETC. The surplus of electrons entering the ETC was passed to AOX (Fig. 2B). It is notable that the RBP6^OE_TbIF1dKO exhibited a markedly elevated oxygen consumption rate on day 2 in comparison to RBP6^OE (231 ± 11 vs. 124 ± 12 pmol/(s*ml), respectively). This was despite the levels of AOX remained comparable between these two cell lines (Fig. 2C, D), suggesting that the loss of TbIF1 may be a contributing factor to this effect. The elevated mitochondrial respiration did not result in discernible changes in ΔΨm at day 2 of RBP6 induction in either cell line as measured in a cell population stained with TMRE (Fig. 2E), but was accompanied by augmented production of mROS, predominantly superoxide, as evidenced by the MitoSOX dye. The RBP6^OE_TbIF1dKO exhibited heightened levels of MitoSOX fluorescence, which aligns with the observed increase in oxygen consumption (Fig. 2F).

In mammalian cells, IF1 plays a role in the stability of ATP synthase type I dimer and oligomer, hence influencing the cristae ultrastructure. Therefore, any alterations in mitochondrial bioenergetics observed in its absence can be attributed to the loss of ATP synthase dimers/oligomers and the subsequent impact on the cristae ultrastructure. To determine whether TbIF1 plays a role in maintaining the stability of ATP synthase type IV dimer in *T. brucei*, we conducted a steady-state analysis of ATP synthase dimers using Blue Native (BN) electrophoresis in RBP6^OE, RBP6^OE_TbIF1dKO, and RBP6^OE_TbIF1^OE. Our findings indicated that there were no discernible differences in the stability of the ATP synthase dimers or any obvious changes in cristae shapes (Supplementary Information Fig. 3).

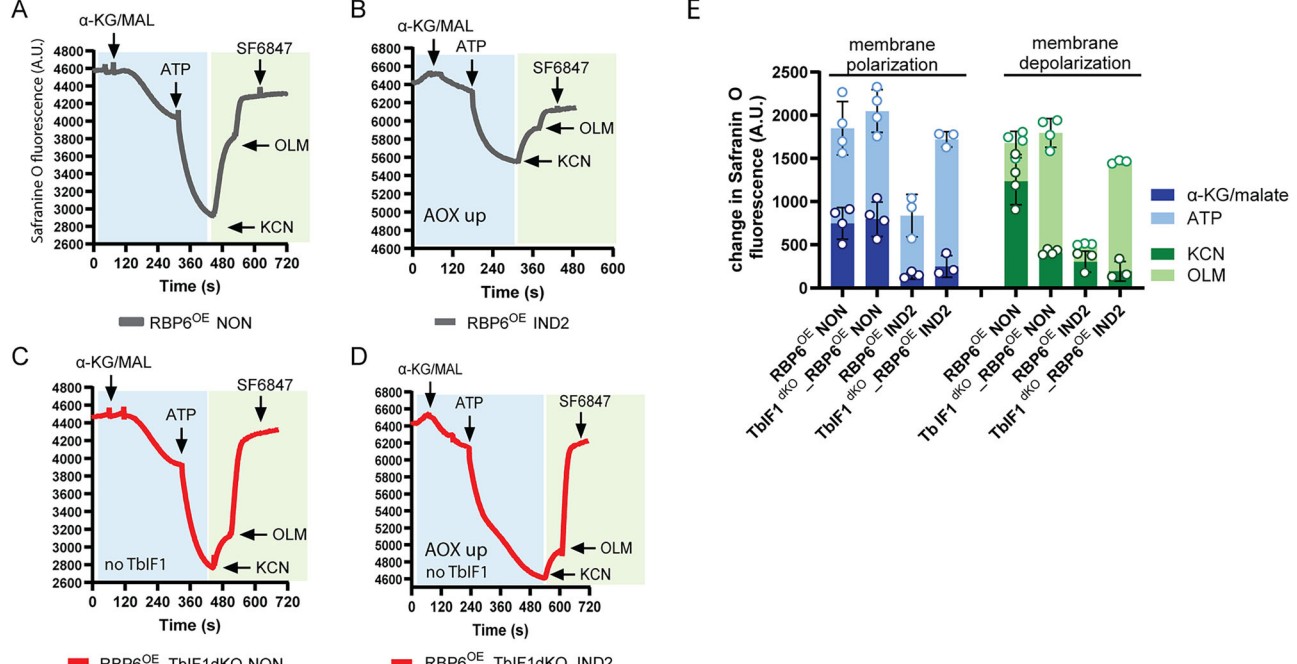

**Fig. 3 | Reversal of *T. brucei* ATP synthase in the absence of TbIF1.**
**A–D** Representative traces of in situ generation and dissipation of the ΔΨm in response to substrates and inhibitors in RBP6[OE] and RBP6[OE]_TbIF1dKO cells induced for 0 and 2 days. Substrates: α-KG/malate (α-KG/MAL), ATP, ADP;
inhibitors: KCN, oligomycin (OLM) and protonophore SF6847. **E** Changes in Safranine O fluorescence after the addition of α-KG/MAL and ATP (membrane polarization, blue columns) and KCN and OLM (membrane depolarization, green columns). (means ± s.d., *n* = 3-4).

## Absence of TbIF1 allows reversal of ATP synthase

We sought to investigate the extent of ATP synthase reversal in RBP6[OE] and RBP6[OE]_TbIF1dKO cells at the 0 and 2 days after RBP6 induction. To this end, an in vitro assay employing digitonin-permeabilized cells and the dye Safranine O was utilized. The Safranine O fluorescence quenching is an indicator of ΔΨm establishment. NADH-producing substrates (α-keto-glutarate/malate) were added in order to create conditions under which electrons could enter the ETC via alternative dehydrogenase, complex I or complex II, and generate ΔΨm via complexes III/IV that is KCN sensitive. Then ATP was added to facilitate ATP synthase reversal. In RBP6[OE] non-induced cells, α-ketoglutarate/malate resulted in a certain degree of fluor-escence quenching of safranine O, and the addition of ATP led to a further increase in ΔΨm. A proportion of this membrane polarization was oligo-mycin-sensitive, indicating that ATP synthase reversal may be a con-tributing factor (Fig. 3A). At day 2 after RBP6 induction, when AOX expression is significantly increased (Fig. 2B), the NADH-producing sub-strates induced lower polarization, suggesting that electrons are partially diverted to non-proton-pumping AOX (Fig. 3B). In RBP6[OE]_TbIFdKO noninduced cells, both substrates induced similar polarization as in RBP6[OE] noninduced, but in the absence of TbIF1, the proportion of polarization generated by ATP synthase was slightly greater (Fig. 3C and E). Finally, in RBP6[OE]_TbIF1dKO induced for 2 days, the NADH-producing substrates induced similar low polarization as in RBP6[OE] induced cells, but the ATP-induced polarization was significantly greater and almost fully abolished by oligomycin (Fig. 3D, E).

The results show that in the presence of the external amount of ATP, the ATP synthase is able to reverse and contribute to the ΔΨm, in addition to the canonical ETC. This in vitro observed phenomenon is even more pronounced in the presence of AOX, which causes lower membrane polarization, and in the absence of TbIF1, which allows full ATP synthase reversal. This is consistent with recently published data showing that there are distinct populations of ATP synthase in mitochondria that can function either synthetically or hydrolytically, depending on the local pmf[61]. The reciprocal relationship between these activities then determines the bioe-nergetic outcome of the whole mitochondrion.

## The reversed activity of ATP synthase contributes to the ΔΨm during the differentiation

Our results demonstrate that ATP synthase in digitonin-permeabilized parasites is capable of efficient reversal and contributes to ΔΨm under conditions of high AOX expression and ATP excess. It would be of interest to determine whether this phenomenon occurs under in living cells undergoing differentiation. To address this question, differentiation was triggered by tetracycline in RBP6[OE]_TbIF1[OE] cell lines, in which the increased expression of TbIF1 is also triggered (Fig. 1A). The ΔΨm was measured in the cell population by flow cytometry using the TMRE dye. The results demonstrated a reduction in TMRE fluorescence on the second day following RBP6 induction (Fig. 4A, B). Since AOX is detectable in the Western blot on this day (Fig. 1D, right panel), it can be concluded that the presence of TbIF1 does not allow the reversal of ATP synthase to partially restore the ΔΨm dissipated by the presence of AOX. The RBP6 induction resulted in slightly higher oxygen consumption (Fig. 4C) and higher mitochondrial ROS levels (Fig. 4D), but to a much lesser extent, suggesting that the presence of TbIF1 interferes with the RBP6-induced remodeling.

Taken together, our data reveal that the programmed downregulation of TbIF1 expression during the RBP6-induced differentiation is associated with increased ATP synthase reversal and metabolic changes characteristic of this developmental transition.

## AMPK is activated during the RBP6[OE] differentiation

Having demonstrated that the presence of AOX and the absence of TbIF1 facilitate partial reversal of ATP synthase to maintain ΔΨm upon ATP consumption, we sought to further investigate whether the cellular ATP levels and ADP/ATP ratio are altered. The ADP/ATP ratio is monitored by the cell in a sensitive manner, as an elevated value indicates an energy crisis that leads to the activation of a number of signaling pathways, including the activation of AMP activated kinase (AMPK). Indeed, while total ATP levels remained unaltered, the ADP/ATP ratio exhibited a statistically significant elevation from day 1 following RBP6 induction in the RBP6[OE]_TbIF1dKO (Fig. 5A, B). The use of an anti-phospho-Thr-172 antibody, which recog-nizes both TbAMPK α subunits, enabled the detection of AMPK subunit α1

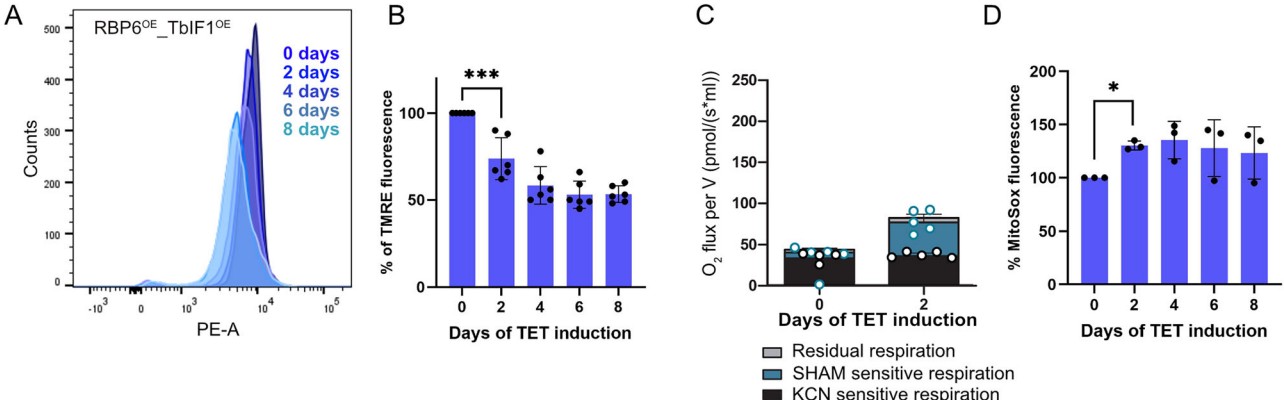

**Fig. 4 | TbIF1 overexpression in RBP6^OE cells leads to lowered ΔΨm. A** A representative flow cytometry histogram of TMRE-stained RBP6^OE_TbIF1^OE cells induced for 0, 2, 4, 6, and 8 days. **B** Flow cytometry analysis of TMRE-stained RBP6^OE_TbIF1^OE cells induced for 0, 2, 4, 6, and 8 days. Individual values shown as dots. (means ± s.d., $n = 5$, *** $P < 0.001$) **C** Oxygen consumption rates in the presence of 5 mM proline in intact RBP6^OE_TbIF1^OE cells induced for 0 and 2 days. The ratio of complex IV- and AOX-mediated respiration was determined using KCN, a potent inhibitor of complex IV, and SHAM, a potent inhibitor of AOX. ($n = 3$) **D** Flow cytometry analysis of MitoSox treated cells to detect mROS levels. Individual values shown as dots. (means ± s.d., $n = 3$, *$P < 0.05$).

phosphorylation, an indicator of AMPK activation[62], in both cell lines, albeit at an earlier time point in RBP6^OE_TbIF1dKO. In contrast, no evidence of AMPK subunit α1 phosphorylation was observed in RBP6^OE_TbIF1^OE cells that did not undergo differentiation into metacyclic cells (Fig. 5C). These findings indicate that AMPK becomes phosphorylated as the parasite differentiates into quiescent metacyclic parasites.

In addition to alterations in the ADP/ATP ratio, AMPK kinase can also be triggered by elevated levels of ROS within cells[63,64]. It is notable that, in contrast to the augmented cellular ROS levels observed in RBP6^OE cells, a considerably greater elevation was detected in RBP6^OE_TbIF1dKO cells, which progress through differentiation into metacyclics more efficiently. In contrast, no increase in cellular ROS levels was observed in RBP6^OE_TbIF^OE cells, whose differentiation is significantly impaired (Fig. 5D).

### The absence of TbIF1 allows the differentiation from metacyclic cells into the long slender bloodstream form

During the differentiation process, we observed a progressive silencing of TbIF1 expression, a change that is essential for establishing and maintaining infection in the mammalian host, as bloodstream form parasites rely exclusively on the hydrolytic activity of the ATP synthase, and TbIF1 presence is lethal to them[45].

In order to fully close the *T. brucei* cycle in vitro, we attempted to differentiate metacyclics obtained by RBP6 induction from the parental line (RBP6^OE) and from the RBP6^OE_TbIF1dKO cell line in vitro. The purified metacyclic form parasites were placed in differentiation medium[65] and subsequently transferred to the HMI-11 medium developed for bloodstream form[66]. Despite repeated attempts, the RBP6^OE-generated metacyclics were unable to yield viable culture (Fig. 6A). Nevertheless, the metacyclics derived from the RBP6^OE_TbIF1dKO consistently differentiated to viable bloodstream form culture that respires solely through AOX and has downregulated ETC complexes III and IV, as evidenced by western blot analysis (Fig. 6B, C). The ability to differentiate from metacyclics to BSF in vitro provides an opportunity to study how the parasite's cellular metabolism and ultrastructure are remodeled upon entry into the mammalian environment, a part of the life cycle that remains largely inaccessible to detailed characterization. However, progress in this area is constrained by the small number of experimental systems in which reliable in vitro differentiation can be achieved[67].

In summary, this study indicates that the regulation of TbIF1 expression and the extent of ATP synthase reversal is closely associated with parasite differentiation. ATP hydrolysis by the ATP synthase complex appears to contribute to the maintenance of the ΔΨm under these conditions, particularly in the presence of AOX, which can be linked to local membrane depolarization. In addition, the parasite's differentiation is accompanied by a number of changes in various cellular activities, such as respiration, mROS production, the ADP/ATP ratio, cellular ROS levels, and the phosphorylation of AMPK (Fig. 7).

## Discussion

Since its discovery in 1963[7], IF1 has been extensively studied in various model organisms as a reversible, non-competitive, and unidirectional inhibitor of ATP hydrolysis by ATP synthase[16,68]. Early studies demonstrated that this protein primarily plays a regulatory role by fine-tuning ATP synthase function, preventing ATP hydrolysis when ΔΨm is compromised due to severe or complete oxygen deprivation or mitochondrial dysfunction associated with a damaged ETC[20,24]. Considering IF1's role in pathophysiology, high IF1 expression has also been reported in various tumor cells, where it exerts pro-oncogenic effects and protects cancer cells from hypoxia-induced stress and apoptosis[17,28]. Nevertheless, IF1 also plays a role under physiological conditions, for example, by contributing to metabolic rewiring in T cells[69] or by controlling energy metabolism during osteogenic differentiation of stem cells[70]. Only recently, it was reported that facilitation of the reverse mode of ATP synthase through IF1 downregulation is key for thermogenesis in brown adipose tissue[71]. Here, we investigated the role of IF1 during the in vitro-induced cellular differentiation of *Trypanosoma brucei*. The programmed differentiation of this parasite involves a reversal of ATP synthase activity: insect-stage forms rely on the forward mode of ATP synthase, whereas mammalian-infectious forms depend on the reverse mode of this molecular nanomachine[59]. However, the molecular mechanisms governing this switch remain unknown.

We utilized the ability to induce *T. brucei* differentiation in vitro through the overexpression of the RBP6 protein, a process accompanied by significant remodeling of the OXPHOS machinery, including the down-regulation of TbIF1[54,55]. This observation suggests that modulation of TbIF1 expression may play a crucial role in ensuring proper differentiation. Through the implementation of loss- and gain-of-function experiments, we demonstrated that TbIF1 downregulation is necessary for parasite metacyclogenesis as its overexpression impairs the parasite's ability to differentiate.

A hallmark of RBP6-induced mitochondrial metabolic remodeling is the early upregulation of AOX following RBP6 induction. Due to its high maximal velocity (Vmax) for ubiquinol[72], *T. brucei* AOX alters electron flow within the ETC. This contrasts with the *Ciona intestinalis* AOX, which is commonly used in many human disease models[73], has a low Vmax[74], and typically becomes active only under conditions of ETC impairment[73]. Upon the RBP6 induction and in agreement with the increased expression of

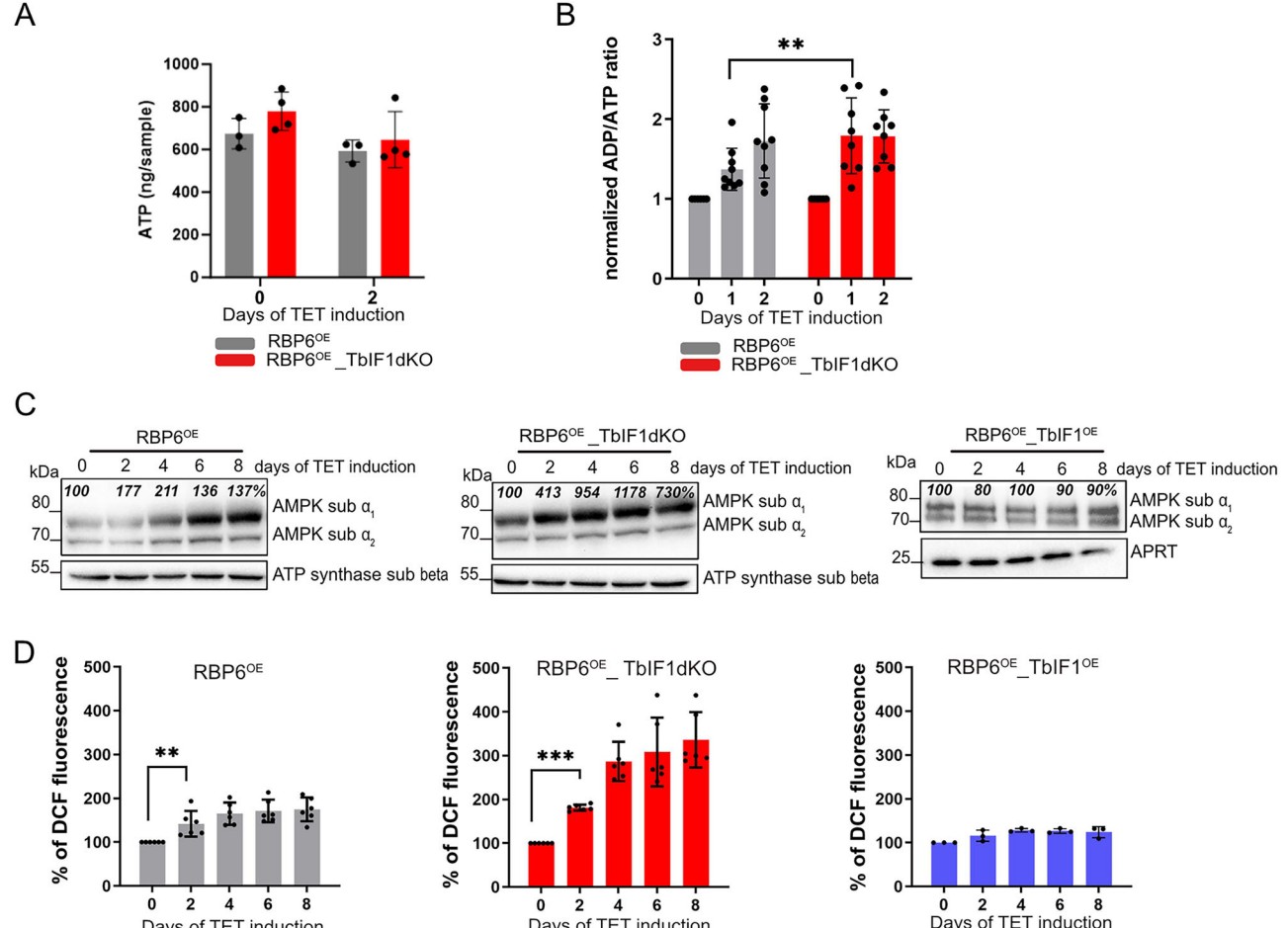

**Fig. 5 | AMPK is activated during *T. brucei* differentiation. A** Steady-state levels of cellular ATP determined by mass spectrometry in intact RBP6[OE] and RBP6[OE]_TbIF1dKO cells. (means ± s.d., *n* = 3–4) **B** Relative ADP/ATP ratios determined in RBP6[OE] and RBP6[OE]_TbIF1dKO expressed as fold increase (means ± s.d., *n* = 6–9, **P* < 0.01). **C** Western blot analysis of whole-cell lysates from RBP6[OE] and RBP6[OE]_TbIF1dKO cells induced for 0, 2, 4, 6, and 8 days using a commercially available anti-phospho-Thr172 AMPK antibody recognizing both AMPK α1 and

α2 subunits. The immunoblot probed with anti-ATP synthase subunit beta and adenosine phosphoribosyl transferase (APRT) antibodies served as loading control. The numbers above the AMPK subunit α1 represent the abundance of immuno-detected protein expressed as a percentage of the noninduced sample after normalizing to the signal intensity of the loading control. **D** Flow cytometry analysis of carboxy-DCF-DA treated cells to detect cellular ROS levels. Individual values shown as dots. (means ± s.d., *n* = 6, **P* < 0.01, ***P* < 0.001).

AOX, NADH-linked substrates induced a lower degree of inner mitochondrial membrane polarization compared to non-differentiating cells. This observation suggests a partial redirection of electrons toward AOX, an enzyme that does not contribute to pmf. Consequently, AOX activity may cause localized depolarization, potentially leading to a reversal of ATP synthase activity within discrete mitochondrial microdomains, despite the overall retention of OXPHOS capacity at the organellar level. Emerging evidence supports the notion that ATP hydrolysis and synthesis can coexist within individual mitochondria, indicating the potential for spatially segregated zones of opposing ATP synthase activity within the same mitochondrial network[29–32]. The extent of the ATP synthase reversal depends on the levels of IF1. In *T. brucei* during the RBP6-induced differentiation, the expression of TbIF1 is strongly downregulated allowing ATP synthase to reverse when ΔΨm is decreased as electrons are partially redirected to non-proton pumping AOX. In addition to the decreased levels of ΔΨm, ATP required for the reversed ATP synthase is produced by mitochondrial substrate-level phosphorylation pathway[50]. During RBP6-induced differentiation, proline oxidation is markedly upregulated, leading to the production of α-ketoglutarate, which serves as a substrate for succinyl-CoA synthetase, an enzyme that generates ATP. This proline-driven respiration is further enhanced in the RBP6[OE]_TbIF1dKO cell line, where TbIF1 is entirely absent. Considering that AOX levels are the same in the RBP6[OE] and RBP6[OE]_TbIF1dKO cell lines, allosteric regulation of AOX by ATP could be

involved in the observed phenomena[75]. The increased proline-driven respiration might also contribute to the elevated mROS levels in the RBP6[OE]_TbIF1dKO cell line, as the reduced NADH molecules must be recycled by enzymes such as complex I or alternative dehydrogenase 2, both known producers of mROS[76].

The importance of ATP synthase reversal during the parasite's differentiation is evident from TbIF1 overexpression experiments, in which RBP6[OE]_TbIF1OE cells exhibited significantly reduced ΔΨm, suggesting that IF1 overexpression impairs the mitochondria's ability to maintain ΔΨm via reverse ATP synthase activity during differentiation. Conversely, during the RBP6-induced differentiation, the TbIF1 expression downregulation is consistent with conditions permissive for ATP synthase reversal, enabling the energetic adaptation required for the metacyclogenesis. The interplay between AOX and IF1 levels, allowing ATP synthase reversal, might be an overlooking phenomenon in studies of various disease etiologies. Xenotopic expression of AOX has been employed as a tool to investigate mitochondrial dysfunction across various disease models, demonstrating notable rescue effects in some cases, while exacerbating the condition in others (reviewed in ref. 73). Given that IF1 expression varies considerably among different cell lines, cell types, and tissues (reviewed in ref. 16), the potential for ATP synthase reversal could be considered when examining disease etiologies.

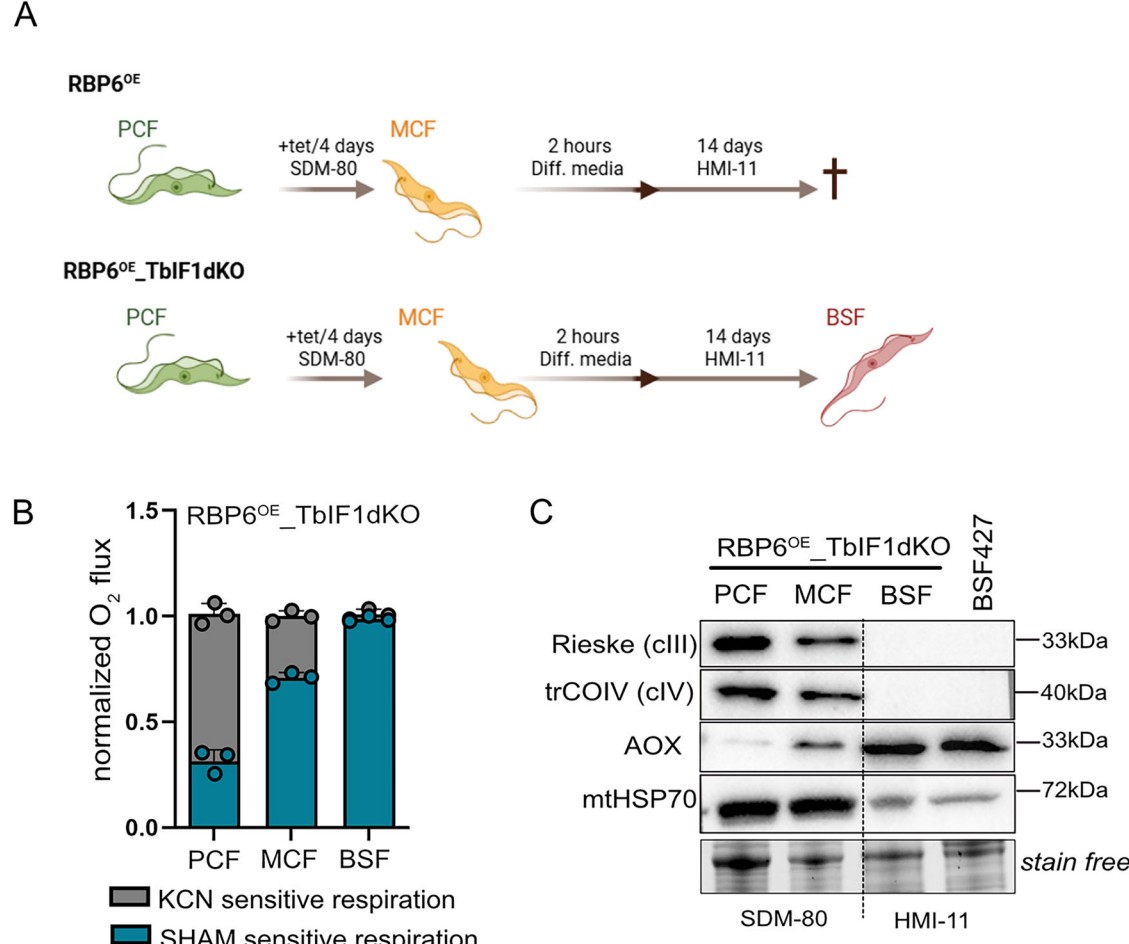

**Fig. 6 | Lack of TbIF1 is an essential prerequisite for successful differentiation to the bloodstream form parasites. A** Scheme of the differentiation protocol. **B** Oxygen consumption rates in the presence of 10 mM glycerol-3-phopshate in intact of RBP6$^{OE}$_TbIF1 dKO procyclic (PCF), metacyclic (MCF), and bloodstream form (BSF) cells measured using O2k-oxygraph. The ratio of complex IV- and AOX-mediated respiration was determined using KCN, a potent inhibitor of complex IV, and SHAM, a potent inhibitor of AOX. Individual values are shown as dots. (mean ±s.d., $n = 3$). **C** Western blot analysis of whole cell lysates from RBP6$^{OE}$_TbIF1 dKO PCF, MCF, BSF cells, and from wild type BSF 427 cells using available antibodies recognizing complex III subunit Rieske, complex IV subunit trCOIV, and AOX. Mitochondrial HSP70 (mtHSP70) serves as a loading control between PCF, MCF, and BSFs samples.

IF1 has traditionally been considered a unidirectional inhibitor of ATP synthase. However, this view has been challenged, as some recent reports have shown that human IF1 might be capable of inhibiting ATP synthase activity in vivo, with this function being regulated by the phosphorylation of serine at the position 14 of the mature human IF1[15,68,77]. Interestingly, this regulatory serine is conserved in humans and mice but is not universally present across the mammalian clade, casting doubt on the generality of this regulatory mechanism[9]. Notably, this serine residue is also absent in *T. brucei*. Consistent with this, our data do not support the hypothesis that TbIF1 also inhibits the forward (synthase) mode of ATP synthase. In *T. brucei* procyclic forms, overexpression of TbIF1 had no effect on ATP production via OXPHOS, nor did it increase the $\Delta\Psi m$[45]. While its function in the procyclic form remains elusive, it likely acts in the canonical manner as an ATPase inhibitor during sudden environmental or cellular changes. Similarly, in *Toxoplasma gondii*, overexpression of IF1 also did not alter ATP synthase activity, ADP/ATP ratio, or $\Delta\Psi m$[78]. These findings suggest that if human IF1 functions also as an inhibitor of the forward mode of ATP synthase, this role may have evolved specifically in certain multicellular organisms, potentially as a part of a more complex regulatory network governing mitohormesis.

Upon mitochondrial depolarization, reversed ATP synthase consumes ATP, initially supplied by the mitochondrial substrate-level phosphorylation pathway. However, if the mitochondrial membrane potential ($\Delta\Psi m$)

falls below a critical threshold, the ATP/ADP carrier also reverses, importing ATP into the mitochondrial matrix[6,79,80]. Therefore, it is expected that the F$_o$F$_1$-ATPase activity influences the cellular ADP/ATP ratio, a critical cellular indicator of the cell's energetic status. Indeed, as early as the first day following RBP6 induction, an increase in the ADP/ATP ratio was observed in both the parental RBP6$^{OE}$ cell line as well as in the RBP6$^{OE}$_TbIF1dKO cells. In general, cells are highly sensitive to fluctuations in the ADP/ATP ratio, as an increase in this ratio signals energetic stress. Elevated ADP/ATP ratios are typically accompanied by increased AMP levels, which are sensed by AMP-activated protein kinase (AMPK), a key regulator of cellular homeostasis[81]. AMPK is responsible for monitoring the energy status of the cell and modulating gene expression to help the cell adapt to reduced ATP levels. During the differentiation of RBP6$^{OE}$ cells, we observed AMPK α1 subunit phosphorylation as an indicator of AMPK activation, with a more pronounced response in the RBP6$^{OE}$_TbIF1dKO line, whereas the phosphorylation was not detected in the RBP6$^{OE}$_TbIF1$^{OE}$ line.

While AMPK activation is typically triggered by elevated AMP levels, there is also evidence that increased ROS levels can contribute to its activation[63,64]. This observation is consistent with the results of our study, which revealed an increase in physiological levels of cellular ROS during differentiation, with higher levels detected in the RBP6$^{OE}$_TbIF1dKO cell line and lower levels in the RBP6$^{OE}$_TbIF1$^{OE}$ cells. *T. brucei* AMPK has been identified as a positive regulator of metacyclogenesis, as RNAi-mediated

**Fig. 7 | TbIF1 is associated with changes in mitochondrial membrane potential and ATP synthase activity during trypanosome differentiation.** Schematic representation of mitochondrial electron transport chain (ETC) function and ATP synthase activity under decreasing levels of TbIF1 expression and increasing steady-state levels of alternative oxidase (AOX) during *Trypanosoma brucei* differentiation. Left: In the microenvironment of the mitochondrial cristae, electrons enter the ETC via complex I, alternative NADH dehydrogenases, or complex II, and flow through complexes III (cIII) and IV (cIV). The resulting mitochondrial membrane potential (Δψm) drives ATP synthesis via ATP synthase. Right: During *T. brucei* differentiation, decreasing levels of TbIF1 and increasing AOX expression are associated with a reduction in Δψm. Under these conditions, ATP synthase appears to operate more frequently in the reverse (ATP-hydrolyzing) mode, which may help to maintain Δψm. The bioenergetic state correlates with an increased ADP/ATP ratio, elevated mitochondrial ROS (mROS), and cytosolic ROS. These changes may together contribute to alterations in cellular signaling pathways, including the activation of AMP-activated protein kinase (AMPK). Created in BioRender. Zíková, A. (2026) https://BioRender.com/fj1hqji.

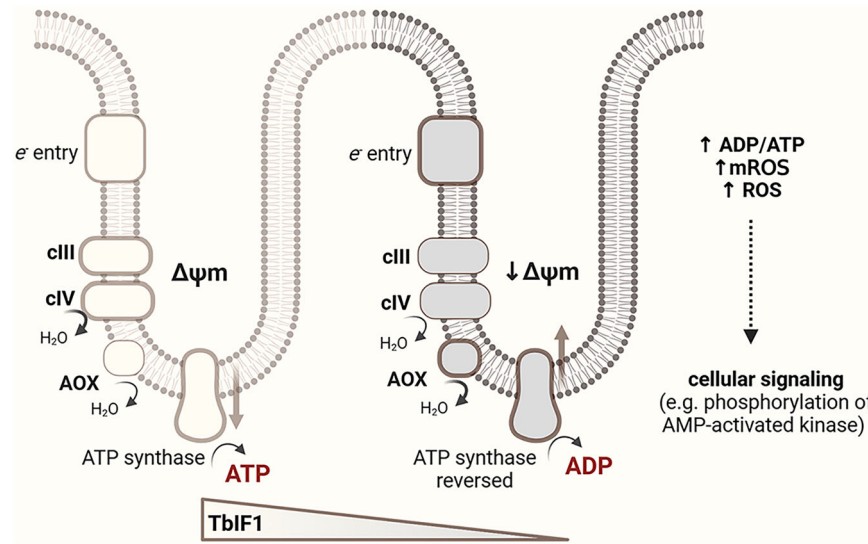

knockdown of all three AMPK subunits significantly reduced the expression of metacyclic-specific genes[57]. Furthermore, AMPK activation is imperative for the differentiation of the proliferative bloodstream form into the quiescent, cell cycle-arrested stumpy form[62]. These stumpy forms are primed for infection of the insect host and exhibit a distinct gene expression profile suited for transmission. A key shared feature of the insect-stage metacyclic form and the bloodstream stumpy forms is that both are non-dividing, cell cycle arrested, exhibit reduced metabolism, and possess a transcriptome prepared for transition to a new host. This suggests that the signaling pathways driving cellular quiescence may be comparable between the differentiation of insect and mammalian forms of the parasite. The involvement of AMPK in parasite differentiation underscores a remarkable evolutionary adaptation, wherein canonical stress-response pathways are repurposed to drive developmental transitions and enhance parasite fitness throughout its complex life cycle[82].

In addition to its established role as a unidirectional inhibitor, IF1 also plays a crucial structural function in mitochondrial cristae organization[24,83]. In mammals, mitochondria are characterized by the presence of lamellar cristae, which feature long rows of ATP synthase dimers arranged along their edges[84]. This configuration generates pronounced membrane curvature that promotes the formation of a cristae environment optimized for OXPHOS. Mammalian ATP synthase typically forms a type I dimer with an angle of approx. 86° between its two central stalks[85,86]. The dimer is characterized by the position of the peripheral stalks, which extend along the longitudinal axis of the dimer. A noteworthy finding is that dimeric IF1, in its active form, spans the interface between adjacent F1 domains of neighboring dimers, forming an inter-dimeric bridge. These tetrameric assemblies have been visualized using cryo-electron microscopy[21,86]. Although the precise biological significance of such inhibited oligomeric complexes remains uncertain, it is plausible that IF1 contributes to their stabilization, thereby supporting cristae integrity. Consistent with this hypothesis, cells lacking IF1 have been reported to exhibit altered cristae morphology[24,26,27,87].

In *Trypanosoma* species, ATP synthase adopts a distinct type IV dimer configuration. This form is characterized by an angle of approximately 60° between the two monomers and the lateral displacement of peripheral stalks to opposite sides of the dimer plane[88]. As a result, dimers positioned at the edges of the discoidal cristae are inclined at a 45° angle relative to the row

axis, forming short helical rows composed of approximately three to six dimers[89]. Structural data from *Euglena* indicate that, unlike the mammalian IF1 which bridges adjacent dimers, the IF1 dimer in these organisms binds individual monomers within the same dimer forming intra-dimeric bridge[90]. The actual dimeric interface between two ATP synthase monomers comprises the subunit e/g module, which is stabilized by bound cardiolipin molecules[88]. The dimer stability seems not to be affected by the absence of TbIF1 as we did not observe changes in ATP synthase oligomerization using BN-PAGE in cells lacking TbIF1. Neither electron microscopy analysis revealed substantial alterations in cristae structure in RBP6[OE]_TbIF1dKO. However, further detailed studies are warranted to assess subtle architectural changes. In *Toxoplasma gondii*, ATP synthase assembles into yet another distinct form, type III dimers, with laterally offset peripheral stalks[91]. In this case, the monomer-monomer angle is approximately 19°, which contrasts with the wider angle observed in type I or IV dimers. Structural data show that dimeric IF1 binds to both monomers within the same dimer, forming an intra-dimeric bridge. Additionally, ATP synthase dimers in *T. gondii* organize into cyclic trimers of dimers. Notably, the dimer-dimer interface involves contacts within the luminal regions and does not incorporate IF1. Nevertheless, knock-out of IF1 in *T. gondii* tachyzoites led to a modest reduction in cristae density, suggesting a potential role for IF1 in maintaining cristae structure[78].

In summary, our study demonstrates that the role of IF1 in controlling ATP synthase activity is more significant in regulating cellular energy metabolism under non-pathological conditions than previously anticipated. In *T. brucei*, the programmed regulation of TbIF1 expression is a critical prerequisite for successful progression through the parasite's life cycle. Our data strongly emphasize the pivotal role of the ATP synthase/IF1 axis in cellular signaling.

## Methods

### Cell lines and culture conditions

The RBP6 overexpression (RBP6[OE]) cell line was generated in ref. 55, with expression induced by daily addition of 10 μg/ml tetracycline. To generate the RBP6[OE]_TbIF1dKO line, both alleles of *TbIF1* (Tb927.10.2970) were sequentially disrupted via homologous recombination. For the first allele, 5′ and 3′ UTRs were PCR-amplified from PCF 427 genomic DNA, cloned into

the pLew13 vector containing a neomycin resistance cassette and a T7 RNA polymerase gene, linearized with NotI, and transfected into PCF 427 cells using AMAXA nucleofection; positive clones were selected with G418. The second allele was disrupted using pLew90 bearing *TbIF1* intergenic homology arms and a hygromycin resistance cassette with a gene for tetracycline repressor, followed by electroporation into single knockout cells and selection with hygromycin[92]. The RBP6 overexpression construct was then introduced into the double-knockout line. To generate the RBP6[OE]_TbIF1[OE] cell line, the *TbIF1* coding sequence was PCR-amplified from PCF 427 genomic DNA and cloned into the pLew79 vector for tetracycline-inducible expression. The construct was linearized with NotI, transfected into RBP6[OE] cells, and puromycin-resistant clones were selected. The PCF 427 (a *T. brucei* strain incapable of completing its development in the tsetse fly) and all the generated cell lines were maintained at 27 °C in glucose-free SDM-80 medium supplemented with 10% heat-inactivated fetal bovine serum (FBS), 7.5 µg/ml hemin, and 50 mM N-acetyl-D-glucosamine to minimize uptake of residual glucose from FBS. Selection antibiotics were included as appropriate to maintain genetic constructs: G418 (15 µg/ml), hygromycin B (25 µg/ml), puromycin (1 µg/ml), and phleomycin (2.5 µg/ml). The bloodstream form was cultured at 37 °C with 5% $CO_2$ in HMI-11 medium supplemented with 10% FBS.

### Cell morphology assessment and fluorescence microscopy
For cell morphology analysis and life cycle stages, $5 \times 10^6$ cells (RBP6[OE], RBP6[OE]_TbIF1 dKO, RBP6[OE]_TbIF1[OE] noninduced and induced) were harvested by centrifugation at $1300 \times g$ for 10 min at room temperature (RT), washed with 1 ml of 1× phosphate-buffered saline (PBS; pH 7.4), and fixed in 3.7% formaldehyde in PBS. Fixed cells were applied to poly-L-lysine-coated coverslips and incubated for 15 min at RT. After three washes with PBS, coverslips were mounted using ProLong Gold Antifade Mountant containing DAPI to visualize nuclear and mitochondrial DNA (kinetoplast). Fluorescence images were acquired using an Axioplan 2 Imaging Universal microscope (Zeiss) equipped with an Olympus DP73 CCD camera. Cell types corresponding to distinct life cycle stages were assigned based on cell size and shape and the relative positioning of the nucleus and kinetoplast. At least 100 cells per time point were scored in a blinded analysis. All experiments were conducted in a minimum of two biological replicates.

### SDS-PAGE and western blotting
Protein samples from whole-cell lysates ($1 \times 10^7$ cells per lane) were separated by SDS-PAGE and transferred to polyvinylidene difluoride (PVDF) membranes. Membranes were probed with appropriate monoclonal or polyclonal primary antibodies, followed by horseradish peroxidase (HRP)-conjugated anti-mouse or anti-rabbit secondary antibodies. Protein bands were visualized using the Pierce enhanced chemiluminescence (ECL) detection system, and signals were captured using a ChemiDoc imaging system (Bio-Rad). Protein sizes were determined by comparison to the PageRuler prestained protein ladder. A commercially available anti–phospho-AMPKα1/2 (Thr172) polyclonal antibody (Sigma-Aldrich) was used at a 1:1000 dilution. All other antibodies were either previously acquired or generated using His-tagged recombinant proteins, with final production performed by Davids Biotechnologie (Regensburg, Germany). Primary antibodies used in this study included: mouse monoclonal anti–mitochondrial HSP70 (1:5000)[93], rabbit polyclonal anti-RBP6 (1:1000, this work), anti-GPEET (1:1000; a generous gift from Prof. Isabel Roditi), anti-BARP (1:1000, a generous gift from Prof. Isabel Roditi), anti-RBP10 (1:1000; this work), mouse monoclonal anti-AOX (1:500; a generous gift from Prof. Minu Chaudhuri), and rabbit polyclonal antibodies against adenosine phosphoribosyl transferase[94] Rieske (1:1000)[55], trCOIV (1:1000)[55], ATP synthase subunit β (1:2000)[95], p18 (1:1000)[95], TbIF1 (1:100)[45], and NDUFA6 (1:1000)[55]. The uncropped and unprocessed Western blot images and stain-free gel scans are presented in Supplementary Fig. 4.

### ADP/ATP ratio
The cellular ADP/ATP ratio was measured using a luciferase-based enzymatic assay kit (Sigma-Aldrich) to assess the energetic status of the cells. For each experiment, $5 \times 10^6$ cells were harvested by centrifugation at $1500 \times g$ for 10 min at RT, then resuspended in 1 ml of 1× PBS; pH 7.4. A total of $1 \times 10^6$ cells per well were transferred into a white 96-well plate, and the assay was performed according to the manufacturer's instructions. Luminescence was measured using a Tecan Infinite M200 plate reader. Each condition was analyzed in both technical and biological triplicates.

### ATP analysis by LC-HRMS
RBP6[OE] and RBP6[OE]_TbIF1 dKO noninduced and induced cells ($5 \times 10^7$) were harvested (1300 g, 10 min, 4 °C), the supernatant removed, and the pellet resuspended in 80 µL cold MeOH:ACN:Water (2:2:1, v/v/v). After 10 min in a 0 °C ultrasonic bath, the mixture was centrifuged (7000 rpm, 10 min, 4 °C). The supernatant was diluted 1:10 with 50% ACN. ^13 $C_{10}$-ATP (20 ng/sample) was added as internal standard (IS) to LC/MS vials, dried under nitrogen, and reconstituted in 40 µL of the diluted extract for LC-HRMS analysis; remaining extract was stored at –80 °C.

ATP quantification followed a liquid chromatography high resolution mass spectrometry (LC-HRMS) method[96] using an Orbitrap QExactive Plus with a Dionex Ultimate 3000 and open autosampler (Thermo Fisher Scientific). The QExactive operated in negative ESI mode (full MS scan, 70–1000 Da) at 70,000 resolution, $3 \times 10^6$ AGC, and 100 ms injection time. Ionization settings included ±3000 V spray voltage, 350 °C capillary/probe temperatures, sheath/auxiliary/spare gases at 60/20/1 au, and S-lens at 60 au. Chromatographic separation was performed on a SeQuant ZIC-pHILIC column (150×4.6 mm, 5 µm, Merck) at 35 °C, 450 µL/min, 5 µL injection. Mobile phase: acetonitrile (A) and 20 mmol/L ammonium carbonate (B, pH 9.2), with gradient: 0 min, 20% B; 20 min, 80% B; 20.1 min, 95% B; 23.3 min, 95% B; 23.4 min, 20% B; 30 min, 20% B. Data were acquired using Xcalibur v4.0. Each condition was analyzed at least in biological triplicate.

### Measurement of cellular and mitochondrial ROS and mitochondrial membrane potential (ΔΨm)
Cellular and mitochondrial ROS levels were assessed using carboxy-2′,7′-dichlorofluorescein diacetate (DCF-DA) and MitoSOX Red, respectively. A total of $1 \times 10^7$ cells were incubated under standard cultivation conditions (27 °C, shaking) with 10 µM DCF-DA (Sigma-Aldrich) for detection of general cellular ROS, or with 5 µM MitoSOX Red Mitochondrial Superoxide Indicator (Thermo Fisher Scientific) for detection of mitochondrial superoxide. After 30 minutes incubation, cells were harvested by centrifugation at $1500 \times g$ for 10 min at RT, washed once with 1 ml of 1× PBS; pH 7.4, and resuspended in 2 ml of PBS. Fluorescence from 10,000 events per sample was measured immediately using a BD FACS Canto II flow cytometer (BD Biosciences).

To assess ΔΨm, cells were stained with 60 nM tetramethylrhodamine ethyl ester (TMRE; Thermo Fisher Scientific) under cultivation conditions for 30 minutes. As a control for mitochondrial depolarization, cells were treated with 20 µM of the protonophore carbonyl cyanide-p-trifluoromethoxyphenylhydrazone (FCCP). Flow cytometry data were analyzed using FlowJo software version 10 (BD Biosciences).

### Measurement of oxygen consumption by high-resolution respirometry
Cellular oxygen consumption was measured using an O2k high-resolution respirometer (Oroboros Instruments) in MiRO5 respiration medium at 27 °C. Each chamber was loaded with $2 \times 10^7$ cells. Mitochondrial respiration was stimulated by the addition of either 5 mM proline or 10 mM glycerol-3-phosphate. To differentiate between cytochrome c oxidase (complex IV)–mediated and alternative oxidase (AOX)–mediated respiration, cells were sequentially treated with 1 mM potassium cyanide (KCN) and 250 µM salicylhydroxamic acid (SHAM), respectively.

### In vitro differentiation of metacyclic form to bloodstream form

To achieve full differentiation from *T. brucei* metacyclic to bloodstream-form, a rodent-free in vitro protocol adapted from cyclical transmission models was used[65]. Briefly, a four-day tetracycline-induced RBP6[OE] and RBP6[OE]_TbIF1dKO cultures containing metacyclic forms was centrifuged at $1300 \times g$ for 10 min at RT and resuspended at a density of $5 \times 10^5$ cells/ml in differentiation medium composed of Minimal Essential Medium with Earle's salts, supplemented with 1% non-essential amino acids, 2 g/l glucose, and 15% heat-inactivated rabbit serum (Sigma-Aldrich). Cells were incubated in this medium for 3 h at 37°C in a closed-lid environment. Following incubation, cells were pelleted again ($1,300 \times g$, 10 minutes, RT) and resuspended in HMI-11 medium supplemented with 1.1% methylcellulose and 10% FBS. Cultures were maintained at 37°C with 5% $CO_2$ for up to two weeks. Upon the emergence of dividing long slender bloodstream forms, the culture was diluted into HMI-11 medium containing 10% FBS.

### Blue Native PAGE and Immunoblotting of $F_0F_1$ ATP synthase

Blue native polyacrylamide gel electrophoresis (BN-PAGE) followed by western blotting was performed to separate protein complexes in their native conformation, using a protocol adapted from[48]. Mitochondria were isolated from $3 \times 10^8$ cells by hypotonic lysis and resuspended in solubilization buffer containing 750 mM aminocaproic acid, 50 mM Bis-Tris, 0.5 mM EDTA (pH 7.0), supplemented with a complete EDTA-free protease inhibitor cocktail and 2% n-dodecyl-β-D-maltoside (DDM). The mixture was incubated on ice for 1 hour to ensure membrane solubilization. Following centrifugation at $16,000 \times g$ for 30 min at 4 °C, the supernatant was collected and protein concentration was determined using the BCA assay. A total of 8 μg of protein was mixed with a Coomassie-based loading buffer (50 mM aminocaproic acid, 0.5% [w/v] Coomassie Brilliant Blue G-250) and loaded onto 3–12% Bis-Tris Native PAGE gels (Thermo Fisher Scientific). Electrophoresis was carried out at 150 V for 3 h at 4 °C. Proteins were transferred to PVDF membranes and probed with specific antibodies as described above.

### Transmission electron microscopy for mitochondrial ultrastructure analysis

Total of $1 \times 10^6$ cells were harvested by centrifugation at $620 \times g$ for 10 min at RT and fixed in 2.5% glutaraldehyde in 0.1 M phosphate buffer (pH 7.2). Post-fixation was carried out with 2% osmium tetroxide for 2 h at 4 °C. Samples were then washed and dehydrated through a graded acetone series and embedded in Polybed 812 resin (Polysciences, Inc.). Ultrathin sections were prepared using a Leica UCT ultramicrotome (Leica Microsystems), and images were acquired with Transmission Electron Microscope JEM 1400 Flash (JEOL) equipped with a XAROSA camera (SIS).

### Statistics and reproducibility

Statistical analyses were performed using GraphPad Prism 10.5.0. Data are presented as mean ± standard deviation (s.d.), and statistical significance was assessed using Student's unpaired t-test, as indicated in the figure legends. Sample sizes (*n*) correspond to independent biological replicates unless otherwise stated. Most experiments were performed with at least three independent biological replicates. For cell morphology analysis, ≥100 cells per time point were scored in a blinded manner. Experiments lacking statistical analysis (e.g., representative western blots, BN-PAGE, and microscopy) were repeated at least twice with similar results.

### Reporting summary

Further information on research design is available in the Nature Portfolio Reporting Summary linked to this article.

### Data availability

All data supporting the findings of this study, including the uncropped and unprocessed Western blot images (Supplementary Fig. 4), are available within the paper. The source data for all charts/graphs, description of *T. brucei* strains, and list of used oligonucleotides can be found in Supplementary Data 1. All other data are available from the corresponding author on reasonable request.

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

## Acknowledgements

We would like to thank Martina Slapničková for excellent technical support and Prof. Christos Chinopoulos (Semmelweis University, Budapest) for stimulating discussions. We would also like to express our gratitude to the Biology Centre core facilities, namely to the Laboratory of Electron Microscopy, to the Laboratory of Microscopy and Histology, and to the Laboratory of Analytical Biochemistry and Metabolomics. This work was supported by the Horizon Europe ERC MitoSignal project no. 101044951, OP JAK CZ.02.01.01/00/22_008/0004575 RNA for therapy, Co-Funded by the European Union and by Czech Science Foundation project no.23-07370S to AZ. We acknowledge the BC CAS core facility LEM supported by the Czech-BioImaging large RI project (LM2023050 and OP VVV CZ.02.1.01/0.0/0.0/18_046/0016045 funded by MEYS CR) for their support with obtaining scientific data presented in this paper.

## Author contributions

M.K. and E.D. performed the experiments and analyzed the data. M.M. performed the mass spectrometry analyses. B.P. contributed to methodology design. A.Z. conceived and supervised the study and acquired funding. A.Z. wrote the first draft of the manuscript, and M.K. and E.D. contributed to subsequent versions and revisions. All authors reviewed and approved the final manuscript.

## Competing interests

The authors declare no competing interests.
