## [Transparent Peer Review file · Communications Biology]

Reversal of ATP synthase is a key attribute accompanying cellular differentiation of *Trypanosoma brucei* insect forms

Corresponding Author: Dr Alena Zíková

Version 0:

Reviewer comments:

Reviewer #1

(Remarks to the Author)

Authors in this manuscript demonstrated that forced downregulation of TbIF1, an endogenous inhibitor of mitochondrial ATPase, promotes RBP6-induced in vitro differentiation of the procyclic to the metacyclic form of *Trypanosoma brucei*, whereas overexpression of TbIF1 inhibits this process. Furthermore, upon RBP6-induction in the absence of TbIF1, the procyclic form completes the differentiation to the bloodstream form under in vitro culture conditions. The results are significant, as this in vitro condition can be used to further investigate the mechanism of differentiation in *T. brucei*. In addition, these studies also show the importance of TbIF1 for parasite differentiation. However, some data were presented poorly, without controls and statistics. Furthermore, in a few cases, results are over-interpreted. All these deficiencies dampen the enthusiasm for the manuscript. Specific points are given below.

1. TbIF1 knockout alone didn't start differentiation unless it is triggered by the induction of RBP6. This fact is not reflected in the abstract. Therefore, the statements in the abstract need to be modified. Furthermore, in vitro differentiation is not equivalent to normal physiological conditions. Thus, 'physiological conditions' should be used cautiously.
2. TbIF1 dKO does not affect procyclic cell growth; it also didn't have much effect on mitochondrial ATP levels and membrane potential. Furthermore, it is toxic to the bloodstream form cells. So, it seems that TbIF1 is expressed in the procyclic form as a checkpoint for differentiation. Is there any other function of TbIF1 in this form?
3. Fig. 1D. WB results were very poorly presented. There are no loading controls, no quantitation. There are 6 lanes for RBP10 in the last panel, whereas others have 5 lanes.
4. Procyclic form should have a low level of AOX. However, WB results in most of the figures showed AOX is undetectable, which is unexpected. Loading controls for some blots are missing.
5. Fig. 2. In RBP6 OE cells, RBP6 levels were induced, but dropped after 12 h. Why?
6. Fig. 2A and C, no loading control, no statistics.
7. Fig. 2E, two graphs need to be consolidated, similar to Fig. 2D
8. Fig. 2 In RBP6 OE-TbIF1 KO cells at Day 2 post-induction, respiration occurs mostly through AOX which should reduce ROS production. Why did the ROS levels increase on day 2? What are the sources of ROS?
9. Although TbAOX (TAO) has higher Vmax and Kcat values, it is not higher than COX [Kido et al., BBA.2009]. Thus, electron flow through AOX without alteration of the AOX protein levels is likely due to its higher Vmax, as the authors mentioned [page 17, line 483-484]. AOX activity is also known to be regulated by ATP levels [Luevano-Martinez et al., FEBS Lett. 2020]. This kind of allosteric modulation could also be involved.
10. AOX upregulation is not robust in RBP6 OE_TbIF1 OE in comparison to that in RBP6 OE and RBP6 OE_TbIF1 dko [Fig. 1D]. What is the reason for these differences?
11. What are the levels of AAC in these cells? Which is important to know. The authors didn't mention that.
12. Fig. 5B. The label for the X-axis needs to be corrected.

Reviewer #2

(Remarks to the Author)

Kunzová et al. genetically manipulate expression of IF1 in trypanosomes by KO and overexpression to show that low level of IF1 stimulates and high level inhibits progression through development from the procyclic to the metacyclic stage in the culture differentiation model driven by overexpression of the RNA-binding protein RBP6. They further show that IF1 deletion enables transition of the metacyclic culture to proliferating bloodstream form. This establishes an essential role of IF1 in the

developmental process and careful bioenergetics experiments provide evidence that low IF1 together with alternative oxidase upregulation effect ATP synthase reversal, known to characterize *T. brucei* bloodstream forms. ATP synthase reversal is accompanied during differentiation by enhanced oxygen consumption, mitochondrial and cellular ROS generation, increase of ADP/ATP ratio and AMPK 1 activation.

This is an interesting study that establishes a physiological role of ATP synthase reversal in a developmental process. Excellent introduction and discussion sections put the observation in context of the large body of literature on IF1 in the mammalian system and make it accessible to a broader readership. The title properly reflects the data reported.

Unfortunately, the authors make some conclusions in the results section on cause – consequence relationships, where only correlations are reported. The phenotypes of IF1 manipulation are very clear, but the relation between ATP synthase reversal, ADP/ATP ratio and AMPK activity is purely correlative. Therefore, suggesting signalling roles of ROS, energy status and AMPK to drive the developmental process is purely speculative and should be moved from results to the discussion. The RBP6 model bypasses the putative physiological cues that trigger development in the tsetse and hence the initiation of the process (“signalling”) remains inaccessible in this experimental context. The respective wording in the results section must be changed.

With respect to understand trypanosome differentiation, there are a few limitations that should be clearly mentioned or discussed: (1) the RBP6 differentiation model is a forced process (not physiologically induced) and the cell populations morphologically differ largely from the stages seen in tsetse infections. (2) the parasite strain used (427) is not able to complete development under physiological conditions in tsetse transmissions. (3) it seems likely that the epimastigotes accumulating at day 2 in culture metabolically correspond to attached (late) epimastigotes in the tsetse as they express AOX. The previous stages dividing epimastigotes, long epimastigotes and short epimastigotes, that are essential for the journey from midgut to the salivary gland and most dependent on energy for strong motility, are not present or distinguishable in the RBP6 model.

Minor comments:

- For the non-expert reader it would be very helpful to make clearer in the introduction and results section that development/differentiation here refers specifically to the steps from procyclic to metacyclic stages, as most trypanosome literature deals with the slender to stumpy and stumpy to procyclic development/differentiation.
- In Fig. 5C the bands are labelled AMPK α 1 Ph and AMPK α 2. “Ph” (for phosphorylation?) is unnecessary as both isoforms are only detected by the antibody when phosphorylated. Using AMPK α 2 as loading control is not acceptable as there is no evidence for absence of regulation; a suitable loading control should be added.
- Fig. 6C uses mt-hsp70 as loading control, which is clearly and strongly regulated. A suitable loading control (e.g. tubulin or PFR) should be added.
- Line 425: the suggestion that AMPK plays a key role in differentiation is not supported and not justified. In the results section no firm conclusions should be made from correlations.
- Line 449: it is not true that this part of the life cycle has not been previously investigated; the authors themselves cite the work from the Tschudi lab for example.

Reviewer #3

(Remarks to the Author)

The manuscript by Kunzová et al on "Reversal of ATP synthase is a key attribute for cellular differentiation of the *Trypanosoma brucei* insect forms" is an interesting study, nicely showing the role of TbIF in differentiation of insect forms and subsequent development of bloodstream forms. It is a valuable tool for the studies in mitochondrial biology. I have only following concerns about the study:

1. Authors refers this study as a "physiological conditions", however overexpression of RBP6 is not anymore physiological condition. So, it is better to avoid the term "physiological".
2. I looked at the expression profiles of TbIF in TritypDB for cultured cells and insect stages. Expression levels in slender and stumpy forms are around 100 TPM, whereas fly midgut forms at day3 TbIF levels are 1000 TPM, highest levels at day 7 >1500 TPM and then gradually declined to 500 TPM in proventricular forms and salivary gland forms. So, in nature, it never goes to zero in salivary gland or bloodstream forms. Authors mention that they have to switch off the expression TbIF in RBP6OE cells for successful differentiation of metacyclic forms into bloodstreams forms. How do they explain this discrepancy to justify that this experimental setup is physiological situation.
3. Figure 1A: TbIF expression is not influenced by RBP6OE, this indicates that RBP6 doesn't have direct role on TbIF. I understand that authors used RBP6 overexpression to induce the procyclic forms to differentiate into salivary gland forms. So, if they Switch off the RBP6 after initial induction, could TbIF KO or TbIF expressing cells (endogenous levels) differentiate into bloodstream forms?
4. Could they give an estimate, how much is overexpressed compared to that of endogenous levels of RBP6 and TbIF?

Version 1:

Reviewer comments:

Reviewer #1

(Remarks to the Author)

The authors have adequately addressed the reviewer's concerns. However, a few problems/comments require further clarification. In addition, a few minor revisions are needed in the manuscript.

1. Fig. 1B. Growth curve: Symbols in the graph are all circles; however, those in the legends are represented by triangles. These should match.
2. Fig. 2B. The unit for the Y-axis has not been explained clearly, neither in the figure legend nor in the method section. What does it mean by Oxygen flux per V (pmole/(s*ml)? it is expected to be like "rate of oxygen consumption/10 to 7 cells."
3. Why was respiration only measured at day 2 and not for subsequent days?
4. COIV levels were reduced significantly at day 2 in RBP6OE and RBP6OE_TbIF1dKO, but KCN-sensitive respiration was not much changed in these cells. How could it be explained?
5. Fig. 2C. It is not clear which band intensities were quantified and shown in the graph. In the corresponding figure legend, the authors said that both RBP6 and AOX band intensities were quantified. However, that is not clearly represented in the graph.
6. Fig. 2E. MitoSox data showed a dramatic increase of ROS at day 2 in RBP6OE_TbIF1dKO but dropped significantly in the following days. Why?
7. Fig. 4C. Y-axis label is missing. AOX levels were comparable in RBP6OE_TbIF1dKO and RBP6OE_TbIF1OE cells at day 2, but the rate of respiration and SHAM-sensitive respiration was not increased in the latter. Thus, OE of TbIF1 inhibits electron flow through ETC. Is that true?
8. Mito ROS was significantly increased in RBP6OE_TbIF1OE cells at day 2, but overall cellular ROS was decreased considerably. How can that be explained?

Reviewer #2

(Remarks to the Author)

The authors have improved the data presentation and the text of this interesting manuscript and responded to all reviewer suggestions.

Minor: During rereading a noticed a typo in the image of Figure 7: AMP activate(d)kinase

Reviewer #3

(Remarks to the Author)

I am satisfied with the revision. Authors have addressed all the concerns.

Version 2:

Reviewer comments:

Reviewer #1

(Remarks to the Author)

Authors' responses to my questions are satisfactory. I enjoyed reviewing the manuscript.

December 12th, 2025

Point-by-point response to the referees' comments

We would like to thank the reviewers for providing fair and constructive comments that helped us to improve the manuscript.

The main changes include the addition of loading controls and statistical analyses. We have also refined the language to be more specific regarding in vitro-induced differentiation, and we avoided using the term “physiological” when referring to this process. In addition, we have modified some of the conclusions to avoid implying cause–consequence relationships when only correlations are reported.

Below, we provide a point-by-point description of the revisions made. Where line numbers are given for the reviewer's reference, they correspond to the manuscript with tracked changes enabled.

Reviewers' comments:

Reviewer #1 (Remarks to the Author):

Authors in this manuscript demonstrated that forced downregulation of TbIF1, an endogenous inhibitor of mitochondrial ATPase, promotes RBP6-induced in vitro differentiation of the procyclic to the metacyclic form of *Trypanosoma brucei*, whereas overexpression of TbIF1 inhibits this process. Furthermore, upon RBP6-induction in the absence of TbIF1, the procyclic form completes the differentiation to the bloodstream form under in vitro culture conditions. The results are significant, as this in vitro condition can be used to further investigate the mechanism of differentiation in *T. brucei*. In addition, these studies also show the importance of TbIF1 for parasite differentiation. However, some data were presented poorly, without controls and statistics. Furthermore, in a few cases, results are over-interpreted. All these deficiencies dampen the enthusiasm for the manuscript. Specific points are given below.

1. TbIF1 knockout alone didn't start differentiation unless it is triggered by the induction of RBP6. This fact is not reflected in the abstract. Therefore, the statements in the abstract need to be modified. Furthermore, in vitro differentiation is not equivalent to normal physiological conditions. Thus, ‘physiological conditions’ should be used cautiously.

We agree with the reviewer.

*The abstract has been revised accordingly and now reads (lines 16–17): “We show that ATP synthase reversal also occurs during in vitro-induced differentiation of the unicellular parasite *Trypanosoma brucei*, partially mirroring events in the tsetse fly.”*

2. TbIF1 dKO does not affect procyclic cell growth; it also didn't have much effect on mitochondrial ATP levels and membrane potential. Furthermore, it is toxic to the bloodstream form cells. So, it seems that TbIF1 is expressed in the procyclic form as a checkpoint for differentiation. Is there any other function of TbIF1 in this form?

Studies in other systems describe IF1 as a protector of ATP synthase, preventing reversal during short-term stress. We consider it likely that TbIF1 plays a similar role in T. brucei, although this remains to be experimentally confirmed. Given that parasites may have adapted stress-related pathways for programmed differentiation (Quintana et al., 2021), the programmed downregulation of TbIF1 may represent an important component of metabolic remodeling. These points are now discussed in the manuscript (lines 607-608,).

3. Fig.1D. WB results were very poorly presented. There are no loading controls, no quantitation. There are 6 lanes for RBP10 in the last panel, whereas others have 5 lanes.

We have improved the figure substantially.

- *Western blots now include mtHsp70 as a loading control.*
- *We added AAC blots and a longer AOX exposure to demonstrate AOX expression in procyclic cells.*
- *All stain-free gels are now provided in the Supplementary Information as additional loading controls.*
- *The lane inconsistency has been corrected.*

4. Procyclic form should have a low level of AOX. However, WB results in most of the figures showed AOX is undetectable, which is unexpected. Loading controls for some blots are missing.

AOX is indeed expressed in PCF. The apparent absence was due to exposure settings used to prevent oversaturation after the ~10-fold upregulation at day 2 (Doleželová et al., 2020). We now provide a longer exposure demonstrating AOX presence in non-induced cells. In RBP6OE_TbIF1OE cells, overexpression is weaker, making AOX detectable even under shorter exposures.

5. Fig. 2. In RBP6 OE cells, RBP6 levels were induced, but dropped after 12 h. Why?

6. Fig. 2A and C, no loading control, no statistics.

We re-examined our replicate experiments and determined that the original western blot was not the most representative. The apparent decrease in RBP6 levels after 12–24 hours was specific to that blot and did not reflect the overall experimental trend. We have now replaced Figure 2A (anti-RBP6) with a more representative blot based on three independent experiments, which does not show a decrease at 24 hours. We have also added mtHsp70 loading controls to Figures 2A and 2C, and all corresponding stain-free gels are now included in the Supplementary Data. Figure 2C has been quantified, demonstrating that AOX expression is regulated at a similar level in both RBP6OE and RBP6OE_TbIF1 dKO cells, with no statistical differences between day 2 and day 4. Figure 2 and its legend have been updated accordingly.

7. Fig. 2E, two graphs need to be consolidated, similar to Fig. 2D

Done.

8. Fig. 2 In RBP6 OE-TbIF1 KO cells at Day 2 post-induction, respiration occurs mostly through AOX which should reduce ROS production. Why did the ROS levels increase on day 2? What are the sources of ROS?

This is a very interesting question, and it is now addressed in the Discussion (lines 577 - 584). It is true that overexpression of AOX in various cell models typically leads to a decrease in mitochondrial ROS, because in those systems (often involving mitochondrial dysfunction or aging),

the main source of mROS is complex III. This is probably not the case in T. brucei during in vitro-induced differentiation, as complex III steady-state levels are actually downregulated. The most likely candidates for mROS production in this context are complex I and the alternative NADH dehydrogenase (being “upstream” of AOX). Considering the increased proline-based respiration, one could expect a higher electron flux through NADH, which is reoxidized by these two components. Additional candidates may include other NADH-dependent mitochondrial dehydrogenases that can also generate superoxide. However, no experimental evidence exists yet (although this is an ongoing project in the lab), so at this stage these points are discussed only as possible explanations.

9. Although TbAOX (TAO) has higher V_{max} and K_{cat} values, it is not higher than COX [Kido et al., BBA.2009]. Thus, electron flow through AOX without alteration of the AOX protein levels is likely due to its higher V_{max}, as the authors mentioned [page 17, line 483-484]. AOX activity is also known to be regulated by ATP levels [Luevano-Martinez et al., FEBS Lett. 2020]. This kind of allosteric modulation could also be involved.

We thank the reviewer for highlighting this point. AOX inhibition by ATP has been described (Luevano-Martínez et al., 2020). Although we measured only total cellular ATP, we now discuss ATP-dependent AOX regulation as a possible mechanism underlying enhanced respiration in RBP6OE_TbIF1 dKO cells (lines 575–577).

10. AOX upregulation is not robust in RBP6 OE_TbIF1 OE in comparison to that in RBP6 OE and RBP6 OE_TbIF1 dko [Fig. 1D]. What is the reason for these differences?

AOX induction is less pronounced in RBP6OE_TbIF1OE cells. This suggests that AOX gene expression regulation is not solely controlled by RBP6 but likely involves additional molecular mechanisms and regulatory factors.

11. What are the levels of AAC in these cells? Which is important to know. The authors didn't mention that.

We agree that AAC levels are relevant.

Western blots for AAC in all three lines have been added to Figure 1D. The steady-state levels do not change substantially during differentiation, irrespective of TbIF1 status.

12. Fig. 5B. The label for the X-axis needs to be corrected.

Corrected.

Reviewer #2 (Remarks to the Author):

Kunzová et al. genetically manipulate expression of IF1 in trypanosomes by KO and overexpression to show that low level of IF1 stimulates and high level inhibits progression through development from the procyclic to the metacyclic stage in the culture differentiation model driven by overexpression of the RNA-binding protein RBP6. They further show that IF1 deletion enables transition of the metacyclic culture to proliferating bloodstream form. This establishes an essential role of IF1 in the developmental process and careful bioenergetics experiments provide evidence that low IF1 together with alternative oxidase upregulation effect ATP synthase reversal, known to characterize T. brucei bloodstream forms. ATP synthase reversal is accompanied during

differentiation by enhanced oxygen consumption, mitochondrial and cellular ROS generation, increase of ADP/ATP ratio and AMPK α 1 activation.

This is an interesting study that establishes a physiological role of ATP synthase reversal in a developmental process. Excellent introduction and discussion sections put the observation in context of the large body of literature on IF1 in the mammalian system and make it accessible to a broader readership. The title properly reflects the data reported.

Unfortunately, the authors make some conclusions in the results section on cause – consequence relationships, where only correlations are reported. The phenotypes of IF1 manipulation are very clear, but the relation between ATP synthase reversal, ADP/ATP ratio and AMPK activity is purely correlative. Therefore, suggesting signalling roles of ROS, energy status and AMPK to drive the developmental process is purely speculative and should be moved from results to the discussion. The RBP6 model bypasses the putative physiological cues that trigger development in the tsetse and hence the initiation of the process (“signalling”) remains inaccessible in this experimental context. The respective wording in the results section must be changed.

We thank the reviewer for this detailed analysis and agree that the relationship between ATP synthase reversal, the ADP/ATP ratio, and AMPK activity is currently only correlative. We identified the relevant sections in the manuscript and revised them according to the reviewer’s suggestion (Abstract: line 18-25; Results: lines 347 – 349, 389, 440–441, 573 etc).

With respect to understand trypanosome differentiation, there are a few limitations that should be clearly mentioned or discussed:

(1) the RBP6 differentiation model is a forced process (not physiologically induced) and the cell populations morphologically differ largely from the stages seen in tsetse infections.

We agree with the reviewer. In response, we have removed the term “physiological” throughout the manuscript and replaced it with “in vitro–induced differentiation” where appropriate. We also revised the abstract to state explicitly that RBP6-induced differentiation only partially recapitulates the events observed in the tsetse fly.

(2) the parasite strain used (427) is not able to complete development under physiological conditions in tsetse transmissions.

We agree. We have added the following clarification in the text (line 177):

*“The PCF 427 strain (a *T. brucei* lineage incapable of completing its development in the tsetse fly)....”*

(3) it seems likely that the epimastigotes accumulating at day 2 in culture metabolically correspond to attached (late) epimastigotes in the tsetse as they express AOX. The previous stages dividing epimastigotes, long epimastigotes and short epimastigotes, that are essential for the journey from midgut to the salivary gland and most dependent on energy for strong motility, are not present or distinguishable in the RBP6 model.

We fully agree with the reviewer. Although the RBP6 system is a valuable model for studying differentiation, it does not reproduce the full developmental progression observed in vivo. We have now addressed this limitation in the Introduction (lines 136–138):

“However, the in vitro-generated epimastigotes likely correspond only to attached (late) epimastigotes, and this system does not recapitulate the full progression from the midgut to the proventriculus and salivary glands.”

Minor comments:

- For the non-expert reader it would be very helpful to make clearer in the introduction and results section that development/differentiation here refers specifically to the steps from procyclic to metacyclic stages, as most trypanosome literature deals with the slender to stumpy and stumpy to procyclic development/differentiation.

We agree with the reviewer. Throughout the manuscript, we have added clarifications that the study focuses specifically on differentiation between insect-stage forms (i.e., metacyclogenesis from procyclic to metacyclic forms). These clarifications appear in the Abstract (lines 16–17) and Introduction (lines 90-91, 144-147, 152).

- In Fig. 5C the bands are labelled AMPK α 1 Ph and AMPK α 2. “Ph” (for phosphorylation?) is unnecessary as both isoforms are only detected by the antibody when phosphorylated. Using AMPK α 2 as loading control is not acceptable as there is no evidence for absence of regulation; a suitable loading control should be added.

We have removed the “Ph” label, as both AMPK isoforms are detected only in their phosphorylated form with this antibody. In addition, AMPK α 2 is no longer used as a loading control, although published studies show that TbAMPK α 2 is not phosphorylated upon AMPK activation and remains unchanged under conditions that strongly induce TbAMPK α 1 phosphorylation (e.g., treatment with 8-pCPT-2'-O-Me-5'-AMP or AMP analogs; Saldivia et al., 2016, Cell Reports). We have now included new loading control, immunoblots using antibodies against ATP synthase subunit β or adenosine phosphoribosyl transferase. Quantification has been performed, and the values are included in the figure.

- Fig. 6C uses mt-hsp70 as loading control, which is clearly and strongly regulated. A suitable loading control (e.g. tubulin or PFR) should be added.

We agree and apologize for the earlier inaccuracy. mtHsp70 is strongly downregulated in bloodstream forms and cannot serve as a loading control between insect-form and bloodstream-form samples. Its use was intended only for comparing samples within the same life-cycle stage.

We have now added a cropped stain-free gel as the appropriate loading control, and the full gel is provided in Supplementary Data

- Line 425: the suggestion that AMPK plays a key role in differentiation is not supported and not justified. In the results section no firm conclusions should be made from correlations.

We fully agree with the reviewer. We have revised the manuscript to avoid overstating conclusions based solely on correlation. We now state that AMPK phosphorylation is associated with RBP6-induced differentiation, and similar overinterpretations have been softened or removed throughout the manuscript (e.g., lines 478-480, 1116–1117).

- Line 449: it is not true that this part of the life cycle has not been previously investigated; the authors themselves cite the work from the Tschudi lab for example.

We thank the reviewer for pointing this out. The sentence has been rephrased to more accurately convey that both the Tschudi model and our system provide valuable tools for studying a portion of the life cycle that remains incompletely characterized, rather than implying that no prior work exists (lines 481–483).

Reviewer #3 (Remarks to the Author):

The manuscript by Kunzová et al on "Reversal of ATP synthase is a key attribute for cellular differentiation of the *Trypanosoma brucei* insect forms" is an interesting study, nicely showing the role of TbIF in differentiation of insect forms and subsequent development of bloodstream forms. It is a valuable tool for the studies in mitochondrial biology. I have only following concerns about the study:

1. Authors refers this study as a "physiological conditions", however overexpression of RBP6 is not anymore physiological condition. So, it is better to avoid the term "physiological".

We agree with the reviewer. This concern was also raised by the other reviewers. The term "physiological conditions" was originally intended to distinguish our work from studies examining IF1 in pathological contexts. However, we recognize that RBP6 overexpression is not physiological. We have therefore removed the term throughout the manuscript and now describe the system as an in vitro-induced differentiation process that partially mimics the parasite's developmental progression within the insect vector.

2. I looked at the expression profiles of TbIF in TritrypDB for cultured cells and insect stages. Expression levels in slender and stumpy forms are around 100 TPM, whereas fly midgut forms at day3 TbIF levels are 1000 TPM, highest levels at day 7 >1500 TPM and then gradually declined to 500 TPM in proventricular forms and salivary gland forms. So, in nature, it never goes to zero in salivary gland or bloodstream forms. Authors mention that they have to switch off the expression TbIF in RBP6OE cells for successful differentiation of metacyclic forms into bloodstreams forms. How do they explain this discrepancy to justify that this experimental setup is physiological situation.

We appreciate this important point. We note that although TbIF1 transcripts are present in insect forms and detectable in bloodstream forms, TbIF1 protein is not detectable in BSF by western blot or proteomic analyses (our unpublished data; consistent with published datasets). Expression of TbIF1 is likely controlled post-transcriptionally. RBP10, for example, regulates mRNA stability and translation (Mugo and Clayton, 2017), and thus transcript presence does not necessarily indicate protein production. We have revised the manuscript to clarify this distinction (lines 489–492).

3. Figure 1A: TbIF expression is not influenced by RBP6OE, this indicates that RBP6 doesn't have direct role on TbIF. I understand that authors used RBP6 overexpression to induce the procyclic forms to differentiate into salivary gland forms. So, if they Switch off the RBP6 after initial induction, could TbIF KO or TbIF expressing cells (endogenous levels) differentiate into bloodstream forms?

This experiment has not been performed. It would be challenging to interpret because RBP6 induction coordinates the progression from PCF to epimastigotes and subsequently to metacyclic forms. Whether cells would continue differentiating into MCFs after RBP6 withdrawal—and whether such MCFs could then transition into BSFs—likely depends on a complex network of

regulatory events beyond RBP6 alone, as RBP6 affects the expression of hundreds of genes. For these reasons, we acknowledge the reviewer's suggestion but note that addressing it falls outside the scope of the present study.

4. Could they give an estimate, how much is overexpressed compared to that of endogenous levels of RBP6 and TbIF?

We agree this information is useful. Quantification has been performed, and the corresponding numerical values demonstrating the degree of overexpression—particularly the substantial overexpression of TbIF1 relative to endogenous levels—have been added to Figure 1A.

BIOLOGY CENTRE CAS

Institute of Parasitology

address: Branišovská 1160/31, 370 05 České Budějovice, Czech Republic

Editor

Communication Biology

January 30th, 2026

Dear Editor,

Thank you for the opportunity to resubmit our revised manuscript, in which we have addressed the concerns raised by the remaining reviewer. We are grateful to the reviewer for identifying several inaccuracies and for the insightful comments provided.

The revisions consist solely of minor textual adjustments. Below, we present a point-by-point description of the changes made, together with our responses to the reviewer's questions.

Overall, we hope that the revised version will now be suitable for publication in *Communications Biology*.

Sincerely,

Alena Zíková, PhD

on behalf of all authors

Institute of Parasitology, Biology Centre

Czech Academy of Sciences

Ceske Budejovice, Czech Republic

Email: azikova@paru.cas.cz

Reviewers' comments:

Reviewer #1 (Remarks to the Author):

Reviewer 1:

Authors in this manuscript demonstrated that forced downregulation of TbIF1, an endogenous 1. Fig. 1B. Growth curve: Symbols in the graph are all circles; however, those in the legend are represented by triangles. These should match.

Corrected.

2. Fig. 2B. The unit for the Y-axis has not been explained clearly, neither in the figure legend nor in the Methods section. What does “oxygen flux per V (pmol/(s·mL))” mean? It would be expected to be something like “rate of oxygen consumption per 10^7 cells.”

Oxygen flux per volume ($\text{pmol}\cdot\text{s}^{-1}\cdot\text{mL}^{-1}$) represents the rate of oxygen consumption normalized to sample volume, expressed as picomoles of O_2 per second per milliliter. This normalization allows direct comparison of respiratory activity between samples of different volumes. We have added this explanation to the figure legend for clarity (lines 628–629).

3. Why was respiration only measured at day 2 and not at later time points?

Since increased AOX expression is one of the earliest hallmarks of RBP6 induction (lines 342–343), we focused on day 2, which represents an early time point at which the majority of data were collected to investigate initial events leading to enhanced differentiation capacity in the RBP6OE_TbIF DKO cell lines. In addition, day 2 cultures are the most uniform, consisting predominantly of epimastigotes. At later time points, cultures become heterogeneous mixtures of procyclic cells, epimastigotes, and emerging metacyclic forms (Figure 1C), which may confound assays measuring properties of the entire cell population. For this reason, data from later time points are reported only as trends.

4. COIV levels were significantly reduced at day 2 in RBP6OE and RBP6OE_TbIF1dKO cells, but KCN-sensitive respiration was not markedly changed. How can this be explained?

These measurements were performed in intact cells rather than isolated mitochondria. Therefore, respiration is influenced by multiple upstream processes, including proline uptake, substrate oxidation, generation of NADH and succinate, and ATP synthase activity, all of which regulate electron flow through the KCN-sensitive ETC. As commonly observed, the ETC does not operate at its maximal capacity under physiological conditions. This can be demonstrated using isolated mitochondria and uncouplers such as FCCP, which collapse the proton gradient and reveal the maximal (uncoupled) respiratory capacity. Thus, reduced COIV levels do not necessarily translate into proportional changes in respiration measured in intact cells. The use of intact cells for respiration measurements is noted in line 349.

5. Fig. 2C. It is unclear which band intensities were quantified and shown in the graph. The figure legend states that both RBP6 and AOX band intensities were quantified, but this is not clearly reflected.

We thank the reviewer for identifying this inaccuracy. The original figure legend was overly complex and unclear. Only AOX band intensities were quantified in RBP6OE and RBP6OE_TbIF1DKO cells at days 2 and 4 and normalized to the HSP70 loading control. The normalized values were then expressed relative to the AOX signal measured in RBP6 OE cells at day 2, which was set to 1. The figure legend has been corrected accordingly, and the Y-axis description has been clarified.

6. Fig. 2E. MitoSOX data show a strong increase in ROS at day 2 in RBP6OE_TbIF1dKO cells, followed by a significant decrease at later time points. Why?

This is a very interesting question and reflects the reviewer's thoughtful engagement with our work. In redox biology, it is well established that cellular ROS levels are determined by both their production and their removal via detoxification systems (e.g., superoxide dismutase for superoxide). Moreover, when ROS act as signaling molecules, they typically do so via a transient spike, a rapid increase in ROS levels that initiates downstream signaling pathways. Subsequently, cells activate detoxification mechanisms to reduce ROS levels, as prolonged ROS accumulation can be detrimental (for a review please look at DOI: 10.1038/s41580-020-0230-3).

7. Fig. 4C. The Y-axis label is missing. AOX levels are comparable in RBP6OE_TbIF1dKO and RBP6OE_TbIF1OE cells at day 2, but respiration and SHAM-sensitive respiration are not increased in the latter. Does TbIF1 overexpression inhibit electron flow through the ETC?

The Y-axis label has been added, and we thank the reviewer for his/her careful attention. Regarding the second point, we respectfully disagree with the reviewer's interpretation. AOX upregulation at day 2 relative to day 0 differs substantially between the two cell lines. Densitometric analysis of the gels shown in Figure 1D reveals approximately a 30-fold increase in AOX levels in RBP6OE and RBP6OE_TbIF1dKO cells, compared to only a 4.5-fold increase in RBP6OE_TbIF1OE cells. Therefore, the more modest increase in respiration in RBP6OE_TbIF1OE cells is likely a consequence of lower AOX respiration. Additionally, since respiration was measured in intact cells, it is possible that activation of the proline oxidation pathway, a hallmark of RBP6-driven differentiation, was also less pronounced in this line. As this was not directly tested, it is not discussed further in the manuscript.

8. Mitochondrial ROS was significantly increased in RBP6OE_TbIF1OE cells at day 2, whereas overall cellular ROS was markedly decreased. How can this be explained?

This is a very interesting question, and addressing it requires consideration of several factors. First, the fluorescent dyes used to detect ROS are designed to report on different ROS species and distinct cellular compartments. While these probes are not perfect and are relatively promiscuous, MitoSOX preferentially reacts with superoxide within mitochondria, whereas 2',7'-dichlorodihydrofluorescein diacetate (DCF-DA) is a cell-permeable probe widely used to assess overall intracellular ROS levels. Following deacetylation and oxidation by ROS, DCF-DA is converted to the fluorescent compound DCF, providing a general readout of cellular oxidative activity rather than of a specific ROS species.

Second, mitochondria and the cytosol represent two largely independent ROS pools, each equipped with robust detoxification systems. Superoxide is considered membrane-impermeable; therefore,

increased superoxide production within mitochondria does not necessarily lead to elevated ROS levels in the cytosol. Mitochondrial superoxide is efficiently detoxified by mitochondrial superoxide dismutase, followed by peroxidases.

Indeed, in RBP6OE_TbIF1OE cells we did not detect increased cytosolic ROS levels. In contrast, the elevated cellular ROS levels observed in RBP6OE and RBP6OE_TbIF1dKO cells are likely derived from multiple intracellular sources. Regarding mitochondrial ROS signaling, recent studies suggest that even membrane-permeable ROS species such as H₂O₂ do not freely diffuse over long distances. Instead, ROS-mediated signaling is thought to occur via redox relay mechanisms involving redox-sensitive proteins that transmit oxidative signals from mitochondria to other cellular compartments. The cellular ROS detected in RBP6OE and RBP6OE_TbIF1dKO cells therefore likely reflect contributions from several ROS-producing pathways, and given the nonspecific nature of the DCF probe, the precise molecular identity of these ROS species cannot be determined.